# Neural connectome of the ctenophore statocyst

**Kei Jokura[1,2,3,4]\*, Sanja Jasek[5], Lara Niederhaus[5], Pawel Burkhardt[6], Gáspár Jékely[4,5]\***

[1]National Institute for Basic Biology, Okazaki, Japan; [2]The Exploratory Research Center on Life and Living Systems, Okazaki, Japan; [3]Grass Laboratory, Marine Biological Laboratory, Woods Hole, United States; [4]Living Systems Institute, University of Exeter, Exeter, United Kingdom; [5]Centre for Organismal Studies, Heidelberg University, Heidelberg, Germany; [6]Michael Sars Centre, University of Bergen, Bergen, Norway

## eLife Assessment

This **fundamental** work significantly advances our understanding of gravity sensing and orientation behavior in the ctenophore, an animal of major importance in understanding the evolution of nervous systems. Through comprehensive reconstruction with volumetric electron microscopy, and time-lapse imaging of cilia motion, the authors provide **compelling** evidence that the aboral nerve net coordinates the activity of balancer cilia. The resemblance to the ciliomotor circuit in marine annelids provides a fascinating example of how neural circuits may convergently evolve to solve common sensorimotor challenges.

**\*For correspondence:**
jokura@nibb.ac.jp (KJ);
gaspar.jekely@cos.uni-heidelberg.de (GJ)

## Abstract

Ctenophores possess a unique gravity receptor (statocyst) in their aboral organ formed by four clusters of ciliated balancer cells that collectively support a statolith. During reorientation, differential loads on the balancer cilia lead to altered beating of the ciliated comb rows to elicit turns. To study the neural bases of gravity sensing, we used volume electron microscopy to image the aboral organ of the ctenophore *Mnemiopsis leidyi*. We reconstructed 1011 cells, including syncytial neurons that form a nerve net. The syncytial neurons synapse on the balancer cells and also form reciprocal connections with the bridge cells that span the statocyst. High-speed imaging revealed that balancer cilia beat and arrest in a coordinated manner but with differences between the sagittal and tentacular planes of the animal, reflecting nerve-net organization. Our results suggest a coordinating rather than sensory–motor function for the nerve net and inform our understanding of the diversity of nervous-system organization across animals.

## Introduction

Ctenophores, or comb jellies, are gelatinous marine animals that actively swim by beating rows of fused cilia known as comb plates. These coordinated ciliary movements not only generate propulsion but also allow precise control of body orientation and directional changes within the water column. Despite lacking a centralized nervous system, ctenophores exhibit sophisticated and dynamic behavioural patterns and respond to a broad range of external stimuli, suggesting the presence of an evolutionarily unique and functionally integrated sensorimotor system coordinating locomotion and posture. Several components of the ctenophore nervous system have been described, including a subepithelial nerve net (SNN), mesogleal neurons, tentacular nerves, and elements of the nervous system in the aboral organ (*Hernandez-Nicaise, 1984*; *Hertwig, 1880*; *Jager et al., 2011*). These neural structures

are characterized by unique features such as a syncytial architecture (*Burkhardt et al., 2023*) and an extensive repertoire of lineage-specific neuropeptides not found in other metazoans (*Hayakawa et al., 2022*; *Sachkova et al., 2021*). Notably, early genomic and single-cell transcriptomic studies failed to identify classical neuronal markers in ctenophores (*Moroz et al., 2014*; *Sebé-Pedrós et al., 2018*). It was only through the use of pro-neuropeptide-based markers that distinct neural cell types were molecularly identified (*Hayakawa et al., 2022*; *Sachkova et al., 2021*). In parallel, phylogenomic and chromosomal synteny analyses have suggested that ctenophores may represent the sister group to all other animals (*Li et al., 2021*; *Ryan et al., 2013*; *Schultz et al., 2023*; *Whelan et al., 2015*), raising the possibility that their nervous system evolved independently from those of other metazoans (*Moroz, 2015*). This hypothesis positions ctenophores as a key lineage for re-evaluating the origins and diversity of neural systems (*Burkhardt and Jékely, 2021*). However, without a detailed understanding of how ctenophore neurons are spatially organized and interconnected, it remains unclear how their nervous system controls behaviour. Clarifying this relationship between neural architecture and behavioural control is essential to uncovering ctenophore neural function.

To address the lack of structural, circuit-level understanding of the ctenophore nervous system, we focused on a unique sensory structure—the aboral organ. This organ functions as a statocyst, a gravity-sensing structure in which a statolith, composed of lithocytes, rests upon four clusters of mechanosensory balancer cells. The statolith acts as a dense mass that deflects the balancer cilia, while the lithocytes are the specialized cells that produce the statolith material (*Noda and Tamm, 2014*). Each balancer cluster serves as a pacemaker: their ciliary beats initiate excitation waves in the ciliated grooves, which are then transmitted to the comb-plate cilia to regulate body orientation (*Horridge, 1965*; *Tamm, 1980*). A key feature of balancer cilia is their bidirectional, mechanoresponsive control of beat frequency. Depending on the direction of deflection and the animal's geotactic state, balancer cells may either increase or decrease their ciliary activity, indicating a dynamic response system (*Horridge, 1965*; *Tamm, 1982*; *Tamm, 2014a*).

In the natural environment, most ctenophores can freely adjust their posture within the water column and exhibit both positively and negatively gravitactic orientations. This behavioural flexibility is likely supported by the aboral organ, which functions as a gravity sensor and pacemaker for ciliary movement. In this study, we represent ctenophores with their aboral organ facing upwards ('balancer-up' posture), as this configuration facilitates intuitive interpretation of balance-like functions and matches the setup used in high-speed imaging experiments.

Previous studies have implicated membrane potential, intracellular calcium concentration, and synaptic input in modulating the balancer response (*Lowe, 1997*). In particular, deflection-induced excitation can be abolished by calcium-channel inhibitors or calcium-free conditions (*Lowe, 1997*), while high extracellular $K^+$ depolarizes and activates cilia in the absence of mechanical stimuli (*Lowe, 1997*). Local calcium application further suggests that $Ca^{2+}$ influx at the ciliary base may be required for the excitatory response (*Lowe, 1997*). These findings support the hypothesis that ionic currents and potential neural inputs play critical roles in shaping balancer-cell output (*Lowe, 1997*).

Ultrastructural and immunohistochemical studies have revealed that the aboral organ contains diverse sensory and neural components, including a distinct neural structure termed the deep nerve net (*Aronova, 1974*; *Hernandez-Nicaise, 1968*; *Jager et al., 2011*). However, how this deep nerve net regulates balancer or other functions has remained elusive.

To investigate how balancer cilia are modulated within the aboral organ, we combined volume electron microscopy (vEM) and high-speed imaging in the ctenophore *Mnemiopsis leidyi*. Through connectomic reconstruction, we characterized the synaptic architecture of the aboral nerve net, revealing synaptic inputs to the balancer cells. By high-speed ciliary imaging, we uncovered correlated activity patterns in balancer cilia across the four quadrants of the apical organ. Our results suggest that the aboral nerve net functions in coordinating the activity of balancer cilia, rather than fulfilling a feedforward sensory–motor function.

## Results

### Volume EM reconstruction of the *Mnemiopsis* aboral organ

For vEM analysis, we used a 5-day-old cydippid larva of *M. leidyi*. To ensure optimal ultrastructural preservation, the specimen was prepared by high-pressure freezing, followed by freeze substitution

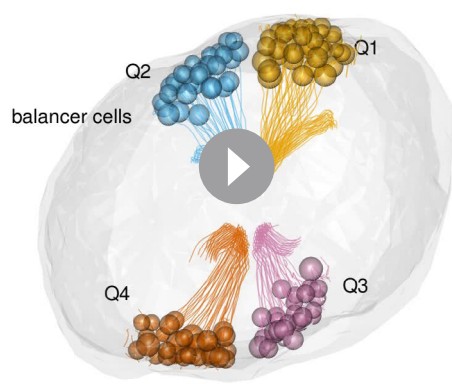

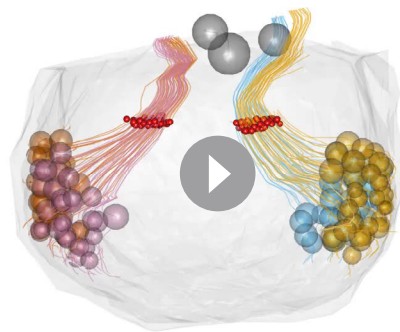

**Video 1.** Morphological rendering of the aboral organ, featuring four distinct cell types—lithocytes, balancer cells, bridge cells, and components of the aboral nerve net (ANN).

https://elifesciences.org/articles/108420/figures#video1

**Video 2.** Visualization of balancer cilia extending from basal bodies. Bundles of balancer cells are shown in different colours. Each red dot indicates the position of a basal body, from which the cilium projects distally. Lithocytes are shown in grey as a reference point, and the outline of the aboral organ is indicated in transparent grey.

https://elifesciences.org/articles/108420/figures#video2

and Epon embedding. From the aboral tip, we collected ~1000 ultra-thin (50 nm) serial sections as ribbons, targeting only the region containing the aboral organ. These were imaged using a Zeiss Gemini 500 SEM at a pixel size (*xy*) of 2.8 nm. We imaged only the region containing the aboral organ. Subsequently, images from 620 layers were stitched and aligned in TrakEM2 (*Cardona et al., 2012*). The final vEM dataset encompassed a volume of 60 μm × 40 μm × 30 μm. We skeletonized and annotated all cells in this dataset in CATMAID (*Saalfeld et al., 2009*; *Schneider-Mizell et al., 2016*). We reconstructed 1011 cells with intact nuclei and somata. Among these, 905 were located within the aboral organ (AO), including balancers and the aboral nerve-net neurons (*Video 1*). Due to the compact and mostly self-contained organization of the AO, these cells were largely contained within the imaged volume. We classified these cells into cell types (*Figure 1—figure supplement 1*, *Figure 1—figure supplement 2*). Our classification was based on (1) ultrastructural features (e.g. number of cilia), (2) cell morphology (e.g. nerve net or bridge cells), (3) unique organelles (e.g. lamellate body, plumose cells), and (4) similarities to cell types previously described by EM. Our classification mostly agrees with the cell types identified in the 1-day-old larva (*Ferraioli et al., 2025*).

In cells containing cilia, we also traced cilia along their length and annotated all basal bodies (*Video 2* and not shown). For neuronal skeletonization, nodes were placed to interconnect the profiles of the same neuron's processes across layers, extending the skeleton until all branches were fully traced. All nodes relevant to synaptic sites were tagged, and skeletons were named and assigned multi-level annotations. As described later, some neurons formed loop-like structures (anastomosed neurons), in which separated branches often rejoined either the main branch or smaller branches (*Burkhardt et al., 2023*). Since CATMAID only supports skeleton trees, branch nodes in such cases were placed near the closest existing node. Seventy-three fragments could not be attached to somata-associated skeletons. Most of these fragments represent short skeletal branches that could not be traced beyond gaps or low-quality layers.

Next, we divided the entire aboral organ into four quadrants (Q1–Q4) to facilitate grouping the identified cells (*Figure 1G*; *Martindale and Henry, 1999*). The general body plan of ctenophores, when viewed from the aboral side, exhibits biradial symmetry around the anal pores (*Martindale and Henry, 1999*). This symmetry corresponds to the four blastomeres present at the four-cell stage during early embryonic development (*Martindale and Henry, 1999*). We assigned 820 of the skeletonized AO cells to one of the four quadrants (*Figure 1G*).

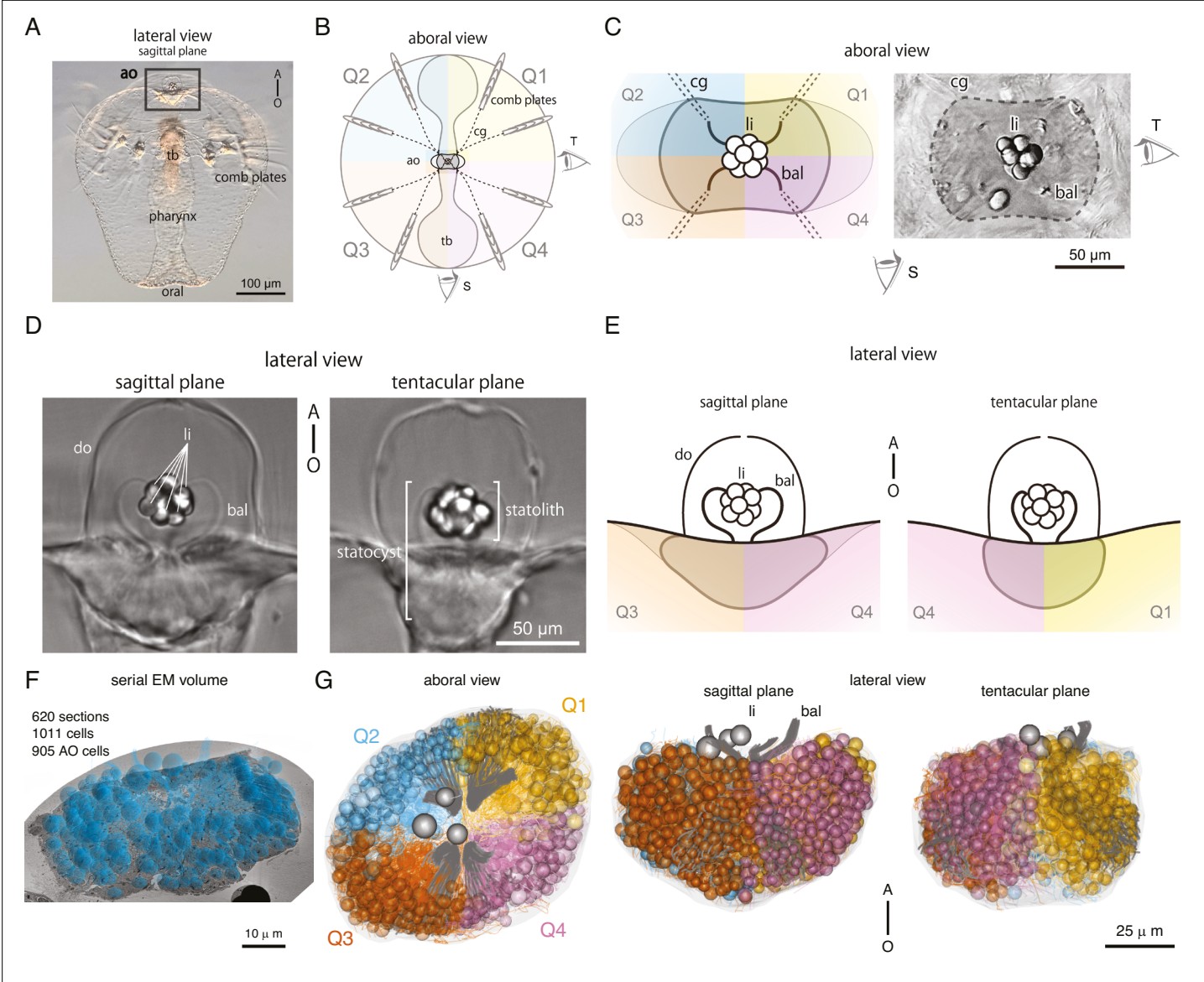

**Figure 1.** Morphology of the aboral organ in *Mnemiopsis leidyi*. (**A**) Whole-body image of a 5-day-old *M. leidyi* cydippid larva viewed in the sagittal plane (lateral view). The boxed region indicates the aboral organ. Abbreviations: ao, aboral organ; tb, tentacle bulb. A–O indicates the aboral–oral body axis. (**B**) Schematic diagram of a cydippid larva in the aboral view. The body is illustrated with four colours and divided into four quadrants, designated as the first (Q1) through fourth (Q4). The two primary viewing angles are referred to as the sagittal plane (S) and the tentacular plane (T). Abbreviations: ao, aboral organ; cg, ciliated groove; tb, tentacle bulb. (**C**) Aboral views of the aboral organ highlighting its spatial organization. The left panel presents a schematic representation of the aboral organ, illustrating the four body quadrants (Q1–Q4), colour-coded consistently with panel (**B**). The right panel shows a corresponding differential interference contrast (DIC) image from the same perspective. The region enclosed by the dotted line delineates the boundary of the aboral organ. The sagittal (S) and tentacular (T) eye icons indicate the viewing directions corresponding to those planes. Abbreviations: bal, balancer; cg, ciliated groove; li, lithocyte. (**D**) Lateral views of the aboral organ in two orthogonal planes captured by DIC microscopy. The left panel shows a view in the sagittal plane, the right panel displays a view in the tentacular plane. The statocyst is a cavity-like organ enclosed by the dome cilia (do), which contains the statolith formed by lithocytes (li) and supported by the balancer cilia (bal). A–O indicates the aboral–oral body axis. (**E**) Schematics of the aboral organ in sagittal (S, left) and tentacular (T, right) planes, corresponding to the views in (**D**). Colours follow the quadrant scheme in (**B**) and (**C**). A–O indicates the aboral–oral body axis. (**F**) Overview of the volume traced in CATMAID with spheres indicating the position of nuclei in the reconstructed cells. Scale bar: 10 μm. (**G**) Morphological rendering of all cells in the aboral organ displayed in the aboral (left panel), sagittal plane (middle panel), and tentacular plane views (right panel). Cells are colour-coded according to quadrants. Lithocytes (li) are represented as three grey spheres, balancers (bal) are depicted as grey lines.

The online version of this article includes the following figure supplement(s) for figure 1:

**Figure supplement 1.** Cell-type composition of the ctenophore aboral organ.

**Figure supplement 2.** 3D reconstruction of all cells comprising the aboral organ.

## An aboral synaptic nerve net of syncytial neurons

Previous immunofluorescent staining in *Pleurobrachia pileus* suggested the presence of a nerve net at the base of the aboral organ, possibly continuous with the surrounding epithelial nerve net (*Jager et al., 2011*). In addition, electron microscopy studies revealed synapses near the floor of the aboral organ, a region where densely packed epithelial cells form a floor-like structure (*Hernandez-Nicaise, 1973*; *Horridge and Mackay, 1964*).

In our volume EM dataset, we identified 396 synapses based on the canonical presynaptic triad morphology: a cluster of synaptic vesicles, a smooth endoplasmic reticulum cisterna, and a mitochondrion. At synaptic sites, we marked mitochondria as synaptic nodes (orange in CATMAID) and connected this node to the nearest node in postsynaptic cells across the regions where synaptic vesicles aligned with the presynaptic cell's membrane, as indicated by light blue arrows in CATMAID (*Figure 2—figure supplement 1A*). No specialized postsynaptic densities were observed, consistent with prior reports (*Hernandez-Nicaise, 1973*). Synapses were either monoadic (one postsynaptic target) or polyadic (multiple postsynaptic targets) (*Figure 2—figure supplement 1B*).

We identified and reconstructed three aboral nerve-net (ANN) neurons, each exhibiting a syncytial morphology characterized by anastomosing membranes and multiple nuclei (ranging from two to five) (*Figure 2A, B*, *Figure 2—figure supplement 1C*, *Video 3*). These three neurons are the only fully reconstructed ANN neurons contained within the volume. Several small ANN-like fragments were also observed at the periphery of the aboral organ, but their continuity with the three ANN neurons remains uncertain.

Near the periphery of the aboral organ, we identified four further anastomosing nerve-net neurons. These resembled the previously reported syncytial SNN neurons in the body wall of *Mnemiopsis* (*Figure 2—figure supplement 1C–G*) and were clearly distinct from the ANN neurons (both in location and morphology).

SNN neurons show a blebbed morphology and contain dense core vesicles (*Burkhardt et al., 2023*; *Figure 2—figure supplement 1C–G*) but no synapses. In contrast, ANN neurons are smooth-surfaced smaller dense vesicles distinct from SNNs and are located centrally within the aboral organ. Furthermore, ANNs are also different from other neurons previously reported by EM, including mesogleal neurons and ciliated sensory cells that synapse on the SNN (types 1–4) (*Burkhardt et al., 2023*). Our results thus identify a second type of syncytial neuron in *Mnemiopsis*.

Based on soma size and spatial distribution, we classified ANN neurons into two types. The first is a single large neuron, ANN_Q1–4, spanning all four quadrants and containing four (possibly five) nuclei. This neuron formed 164 presynaptic sites. The second type includes two smaller neurons, ANN_Q1Q2 and ANN_Q3Q4, each confined to two adjacent quadrants and containing two nuclei. ANN_Q1Q2 had 121 presynapses, while ANN_Q3Q4 had 67 (*Figure 2C–F*). In contrast, SNN neurons exhibited no synaptic contacts.

## Synaptic connectome of the gravisensory organ

Synapse annotation revealed multiple types of connections involving ANN neurons within the gravisensory organ. We identified synapses between different ANN neurons and synapses that ANN_Q1–4 forms on itself (autapses). The ANN neurons also form synapses on the gravity-sensing balancer cells and the bridge cells. Bridge cells, but not balancer cells, also synapse on the ANN.

Balancer cells are monociliated cells with long motile cilia that form four bundles of compound cilia, one in each quadrant (*Video 2*). These bundles come together at the centre of the aboral organ to support the cellular mass of the statolith. In each quadrant, there were between 28 and 37 balancer cells (Q1: 37; Q2: 32; Q3: 32; Q4: 28). Each cell contained three to seven mitochondria. The curvature of the ciliary bundle and the position of somata differed clearly when viewed laterally from the sagittal or the tentacular plane (*Figure 3B*).

ANN_Q1–4 formed synapses on balancer cells in all four quadrants (on 8 cells in Q1, 11 cells in Q2, 8 cells in Q3, and 10 cells in Q4) while ANN_Q1Q2 and ANN_Q3Q4 synapsed to balancer cells in their respective quadrants (ANN_Q1Q2 on 7/37 cells in Q1 and 7/32 cells in Q2; ANN_Q3Q4 on 2/32 cells in Q3 and 2/28 cells in Q4). Some balancer cells received inputs from both ANN_Q1–4 and either of ANN_Q1Q2 or ANN_Q3Q4. Notably, we found no presynaptic sites in balancer cells (*Figure 3—figure supplement 1*), contradicting earlier suggestions of afferent signalling from balancer cells to neurons (*Hernandez-Nicaise, 1974*).

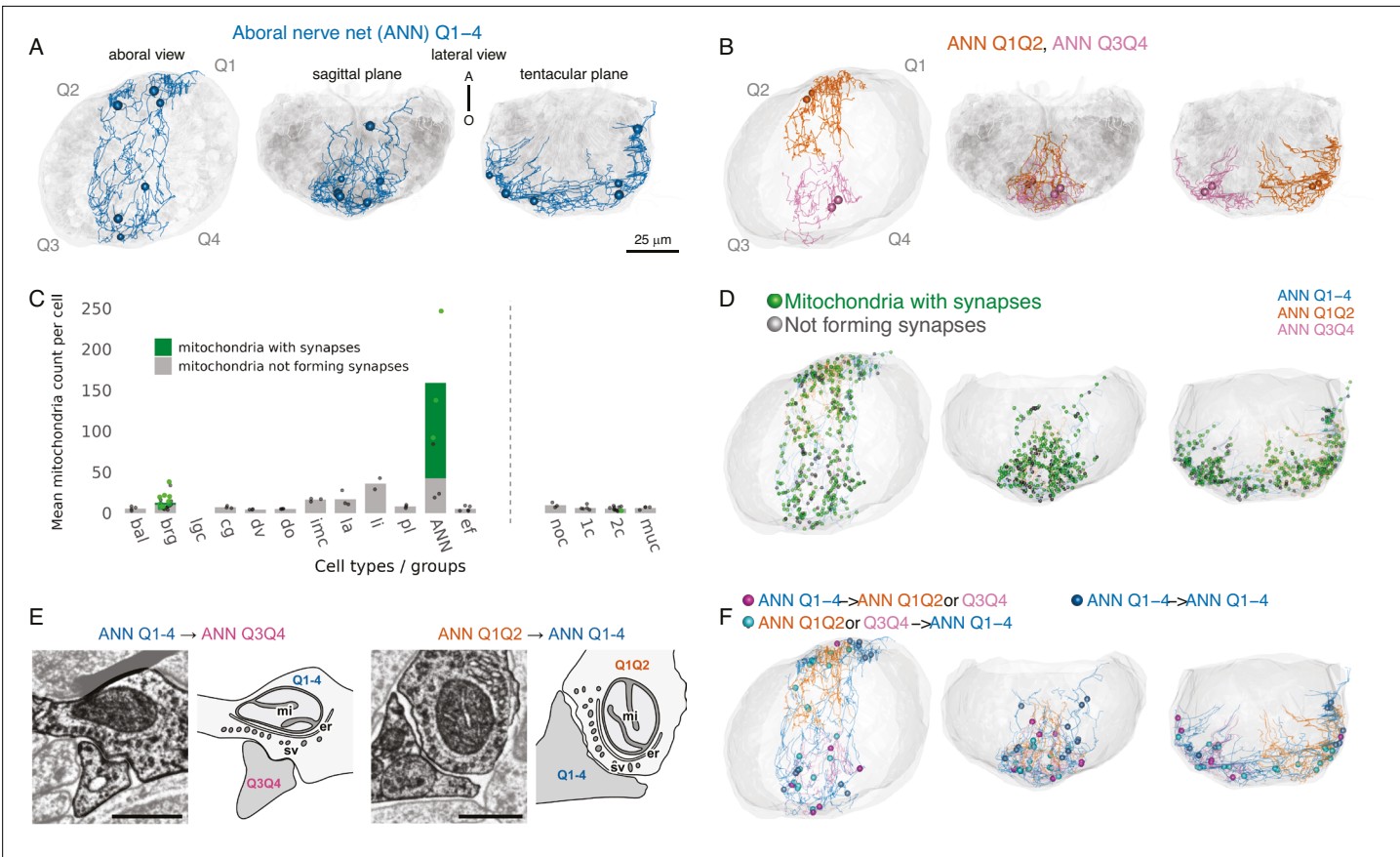

**Figure 2.** Organization of the aboral synaptic nerve net. (**A**) Morphological rendering of the large aboral nerve-net (ANN) neuron ANN_Q1–4 spanning all four quadrants, in aboral (left), sagittal (middle panel), and tentacular view (right panel). Spheres indicate the positions of nuclei. (**B**) Morphological rendering of the two smaller ANN neurons ANN_Q1Q2 (magenta) and ANN_Q3Q4 (orange), each spanning two quadrants, in aboral (left), sagittal (middle panel), and tentacular (right panel). Spheres indicate the positions of nuclei. (**C**) Number of mitochondria per cell for each cell type. The number of synapse-associated mitochondria is shown in green, the number of mitochondria outside synapses is shown in grey. *N* > 2 cells for each cell type except lithocytes (only 2 cells are fully within the volume). Abbreviations: bal, balancer; brg, bridge; lgc, large granular cell; cg, ciliated groove; dv, dense vesicle cells; imc, intra-multiciliated cells; la, lamellate bodies; li, lithocytes; pl, plumose; ANN, aboral nerve net; ef, epithelial floor cells; noc, non-ciliated; 1c, monociliated; 2c, biciliated; muc, multiciliated cells. (**D**) Positions of mitochondria within the ANN neurons. Green marks mitochondria associated with presynaptic triad structures. Black marks mitochondria categorized as 'Not forming synapses', which includes cases where synaptic vesicles were present but without a clear triad, vesicle identification was uncertain, or vesicles were absent. (**E**) Representative electron micrographs of presynaptic triad structures observed in the dataset and their schematic diagrams. The left diagram illustrates synaptic projections from the central ANN (ANN Q1–4) to the lateral ANN (ANN Q3Q4), while the right diagram shows synaptic projections from the lateral ANN (ANN Q1Q2) to the central ANN (ANN Q1–4). Cells other than ANNs have been greyed out in the electron micrographs. Abbreviations: mi, mitochondrion; er, endoplasmic reticulum; sv, synaptic vesicles. Scale bar: 500 nm. (**F**) Position of synapses. Synapses from the central ANN to the lateral ANN are shown in magenta, while synapses from the lateral ANN to the central ANN are shown in cyan. Blue dots indicate the locations of autapses within ANN_Q1–4.

The online version of this article includes the following source data and figure supplement(s) for figure 2:

**Source data 1.** Average numbers of mitochondria that are part of a synapse versus mitochondria that are not part of a synapse, per cell type.

**Source data 2.** Counts of mitochondria that are part of a synapse versus mitochondria that are not part of a synapse, per cell type.

**Figure supplement 1.** Ultrastructural features and synaptic organization of the aboral nerve net (ANN), in comparison to the subepithelial nerve net (SNN).

The second group of cells that formed synaptic contacts with the ANN were the bridge cells. Bridge cells, first described in 2002 (*Tamm and Tamm, 2002*), are characterized by bundles of elongated processes filled with microtubules that arch over the epithelial layer, resembling a bridge. Their somata are located at the base of the paired balancer-cell clusters along the tentacle surface and extend across the sagittal plane towards the base of the opposite balancer cells. Bridge cells form two distinct cell groups across the sagittal plane, in the Q1Q2 and Q3Q4 regions. In the Q1Q2 region, we identified 14 bridge cells, in Q3Q4, 12 cells (*Figure 3—figure supplement 1*, *Video 1*).

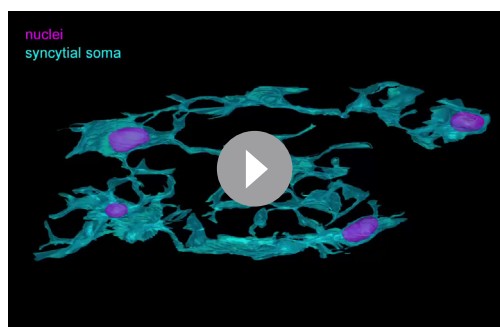

**Video 3.** Volumetric reconstruction of a part of the aboral nerve-net (ANN) Q1–4 neuron showing syncytial soma (cyan) and nuclei (magenta). The rotating view highlights the anastomosing morphology.
https://elifesciences.org/articles/108420/figures#video3

In contrast to balancer cells, bridge cells participated in reciprocal synaptic interactions with ANN neurons, forming both pre- and post-synaptic connections.

Nearly all bridge cells (25/26) received synaptic input from ANNs. Bridge cells in Q1Q2 received inputs from ANN_Q1Q2 (11 cells), ANN_Q1–4 (1 cell), or both (2 cells). In Q3Q4, synapses on bridge cells were from ANN_Q3Q4 (1 cell), ANN_Q1–4 (8 cells), or both (3 cells).

Bridge cells also had presynaptic sites with the typical presynaptic triad structure near 30% of their mitochondria. The presynaptic triads are concentrated on the basal end of the cell, near the nucleus. These bridge-cell synapses were formed on ANN neurons or other bridge cells. Bridge cells in the Q1Q2 region formed synapses on ANN_Q1Q2 (3 cells) or on both ANN_Q1Q2 and ANN_Q1–4 (2 cells). Bridge cells in Q3Q4 synapsed on ANN_Q3Q4 (1 cell) or ANN_Q1–4 (6 cells).

In both the Q1Q2 and the Q3Q4 regions, bridge cells also formed synapses with adjacent bridge cells. However, we found no synapses across the sagittal plane to bridge cells in the opposite region.

To examine the overall synaptic organization of the balancer organ, we grouped cells of the same type and region into network nodes, summed the synaptic counts, and mapped the network onto the four-quadrant anatomy (*Figure 3H*). The resulting connectivity graph revealed feedback loops among ANN neurons and between ANN and bridge cells, displaying biradial symmetry. Local sub-circuits centred around ANN_Q1Q2 and ANN_Q3Q4 were evident, while ANN_Q1–4 formed global connections across quadrants.

Surprisingly, and contrary to the expected sensory–motor or input–output model of balancer function, we found no synapses from the mechanosensory balancer cells to ANN neurons, nor from ANN neurons to known motor effectors such as ciliated groove cells.

The ANN neurons also formed synapses on other cell types in the aboral organ, including the dense-vesicle cells, epithelial floor cells and several non-ciliated, monociliated, or biciliated cells (*Figure 3—figure supplement 1*). None of these other cell types synapsed on the ANN. These cell types and synaptic connections are outside the gravisensory organ and are not considered further here.

## Dynamics of balancer cilia imply a coordinating function for the nerve net

To investigate whether balancer cilia are coordinated by neural inputs, we did high-speed video microscopy recordings to analyse their activity patterns across body axes. Previous studies (*Lowe, 1997*; *Tamm, 1982*; *Tamm, 1980*) have established that balancer cilia function as mechanoreceptors, with their beating frequency modulated by inclination. Tamm also suggested that differences in statolith morphology and the shape of balancer cilia between the tentacular and sagittal planes could lead to different forces exerted on cilia by the statolith (*Tamm, 2015*; *Tamm, 2014b*). To further explore this, we used a tilted microscope with a vertical stage where we mounted immobilized cydippid larvae with their aboral–oral axis aligned in different orientations relative to the gravity vector (*Videos 4–8*). Larvae were oriented with their sagittal or tentacular plane parallel to the sample stage. We could confirm earlier findings that the beating frequency of balancer cilia is dependent on the inclination (*Video 8*). Our aim here was to focus on spontaneous ciliary beating under conditions of equal load from the statolith across quadrants. Therefore, we only analysed larvae whose aboral–oral axis was approximately parallel to the gravity vector (mouth facing downward within a 0–20° deviation). We then compared larvae oriented either with their sagittal or tentacular plane parallel to the sample stage (*Figure 4A*).

In larvae with their sagittal plane facing the objective, we could compare balancer-cilia movements between Q1 vs. Q2 and Q3 vs. Q4. In other larvae oriented in the tentacular plane, we could

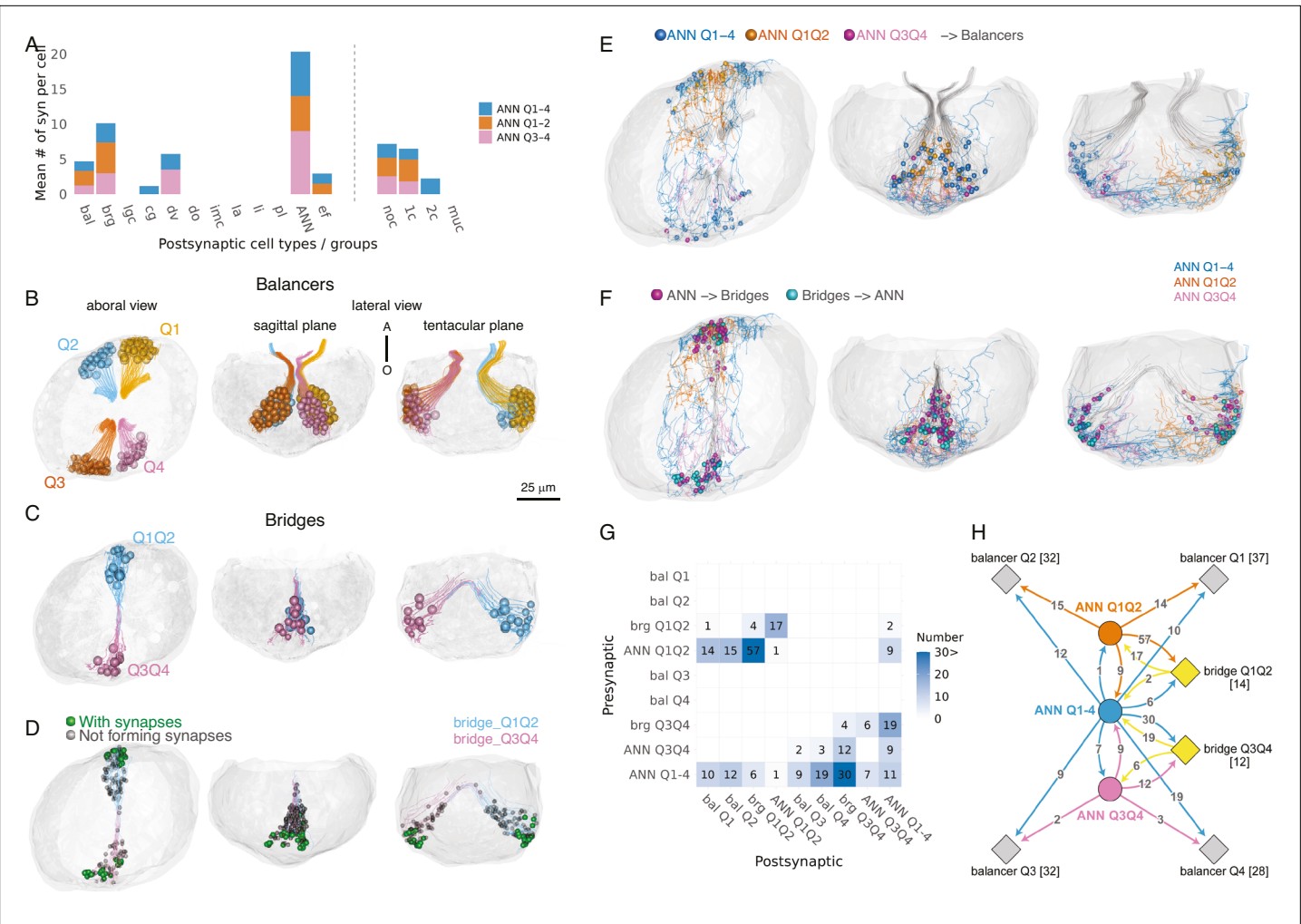

**Figure 3.** Synaptic connectivity of the balancer and bridge cells with the ANN. (**A**) Number of synaptic inputs from the ANNs (ANN_Q1–4: blue, ANN_Q1Q2: orange, ANN_Q3Q4: magenta) to each cell type. Abbreviations: bal, balancer; brg, bridge; lgc, large granular cell; cg, ciliated groove; dv, dense vesicle cells; imc, intra-multiciliated cells; la, lamellate bodies; li, lithocytes; pl, plumose; ANN, aboral nerve net; ef, epithelial floor cells; noc, non-ciliated; 1c, monociliated; 2c, biciliated; muc, multiciliated cells. (**B**) Morphological rendering of balancer cells, with each cell colour-coded by quadrant: Q1 (yellow), Q2 (blue), Q3 (orange), and Q4 (magenta). Each sphere represents the position of an individual nucleus. The fine projections extending from the cells represent traced skeletons that follow the cell body and continue into a single cilium emerging from the soma. Views are shown from three orthogonal perspectives: aboral view (left), sagittal plane (middle), and tentacular plane (right). The spatial arrangement highlights the quadrant-specific organization of balancer clusters within the aboral organ. (**C**) Morphological rendering of bridge cells spanning the Q1Q2 and Q3Q4 quadrants. The Q1Q2-side bridge cells are shown in blue, while the Q3Q4-side bridge cells are shown in magenta. The morphology of individual bridge cells extending across tentacular-axis-symmetric quadrant regions is depicted. The spheres represent the positions of individual nuclei. (**D**) Positions of mitochondria within bridge cells. Green marks mitochondria associated with presynaptic triad structures; black marks mitochondria categorized as 'Not forming synapses' (including cases with vesicles but no clear triad, uncertain vesicles, or no vesicles). (**E**) Synaptic connections from ANN neurons to balancer ciliated cells. The positions of synapses from ANNs to balancers (coloured spheres) are indicated. The three ANN cells and their respective synapses are coloured differently (ANN Q1–4, blue; ANN Q1Q2, orange; ANN Q3Q4, magenta). Balancer ciliated cells are shown in light grey. (**F**) Synaptic connections between ANNs and bridge cells. The positions of synapses from ANNs to bridge cells (magenta spheres) and from bridge cells to ANNs (light blue sphere) are indicated. The skeletons of the three ANN neurons are shown (ANN Q1–4, blue; ANN Q1Q2, orange; ANN Q3Q4, magenta). Bridge cells are shown in light grey. (**G**) Connectivity matrix of the gravity-sensing neural circuit. Columns represent presynaptic cell groups, while rows represent postsynaptic cell groups. The numbers and varying shades of blue correspond to the number of synapses. (**H**) Complete synaptic wiring diagram of the gravity-sensing neural circuit. Cells belonging to the same group are shown as diamonds, with the number of cells shown in square brackets. The number of synapses is shown on the arrows. In panels **B–F**, the left view shows a dorsal view of the aboral organ, the middle panel a sagittal plane view and the right panel a tentacular plane view.

The online version of this article includes the following source data and figure supplement(s) for figure 3:

**Source data 1.** Synapses from the aboral nerve net (ANN) to downstream neurons, annotated by postsynaptic cell type.

*Figure 3 continued on next page*

*Figure 3 continued*

**Source data 2.** Synapse counts between balancer, bridge, and aboral nerve-net (ANN) neuron-groups used to generate the connectivity matrix.

**Figure supplement 1.** Full synaptic connectome of the ctenophore aboral organ.

**Figure supplement 1—source data 1.** Synapse counts between individual aboral organ cells used to generate the full connectome graph.

simultaneously image Q1 and Q4 or Q2 and Q3. We used a high-speed camera and recorded at 100 fps for 2 min (12,000 frames). To analyse ciliary beating, we selected regions of interest (ROIs) in areas where brightness changes indicated ciliary beating. Ciliary beating was both manually quantified and plotted as kymographs.

During the 2-min recordings, balancer cilia could beat fast, slow, or exhibit abrupt stops of beating (arrest) followed by re-initiation (re-beat). Occasionally, large body contractions moved the entire cydippid out of frame, and data from these episodes were excluded. We focused on three metrics for inter-quadrant comparison: (1) the timing of ciliary arrest, (2) the timing of re-beat after an arrested phase, and (3) ciliary beat frequency (CBF).

We found that arrest and re-beat events were synchronized between balancers across the sagittal plane. However, while the timing of re-beat was also near-simultaneous along the tentacular plane, arrest timing was offset by up to 2.16 s (*Figure 4D*, *Videos 4–7*). Overall, these data reveal that arrests are only coordinated between Q1–Q2 and Q3–Q4, whereas re-beat is coordinated over all four quadrants.

As shown in *Figure 4E*; *Figure 4—figure supplement 1*, beat frequencies of left and right balancers fluctuated substantially in both orientations. However, while left–right activity was largely synchronized in sagittal pairs (*Figure 4E*, left), tentacular pairs (*Figure 4E*, right) displayed strikingly periodic alternations between left and right balancer activity. This pattern suggests a consistent temporal offset between the two sides, possibly arising from differences in the timing of arrest or re-beat events.

To quantify this, we calculated a rolling Pearson correlation between left and right balancer activity in each plane. The resulting mean correlations were significantly higher in the tentacular plane compared to the sagittal plane (*Figure 4F*; *t*-test, p = 0.014). This result indicates that tentacular balancers beat in a more coordinated, mirror-symmetric manner, whereas sagittal balancers exhibit greater variability between sides.

Evaluating these data in light of the circuit diagram suggests that shared ANN inputs (ANN_Q1Q2 or ANN_Q3Q4) to a pair of balancers along the sagittal plane may underlie their synchronized arrests. In contrast, in the tentacular plane, separate ANNs innervate the balancer pairs. At the same time, the ANN_Q1–4 neuron synapses on all four balancers, hinting at a neural substrate for their synchronized re-beat.

Notably, bridge neurons associated with each quadrant region (Q1Q2 or Q3Q4) extend their processes across the midline, forming connections that span between opposite quadrant domains

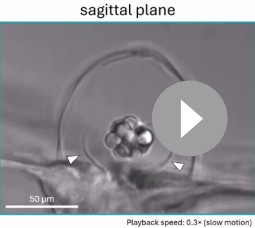 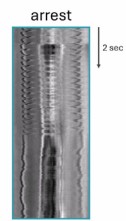 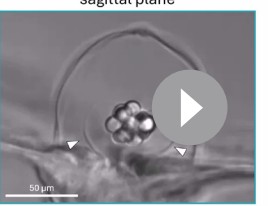 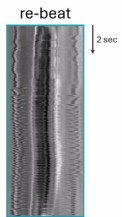

**Video 4.** High-speed imaging of balancer ciliary beating along with kymographs from two individual balancer ciliary bundles. The recording was acquired at 100 frames per second and played back at 30 frames per second (0.3×). The white line overlaid on the kymographs indicates the current time point shown in the video. This example captures a ciliary arrest event in the sagittal plane.

https://elifesciences.org/articles/108420/figures#video4

**Video 5.** High-speed imaging of balancer ciliary beating along with kymographs from two individual balancer ciliary bundles. The recording was acquired at 100 frames per second and played back at 30 frames per second (0.3×). The white line overlaid on the kymographs indicates the current time point shown in the video. This example captures a ciliary re-beat event in the sagittal plane.

https://elifesciences.org/articles/108420/figures#video5

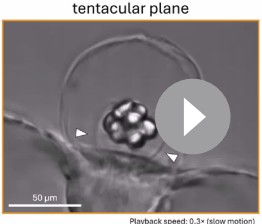
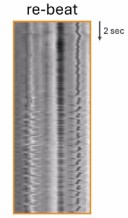
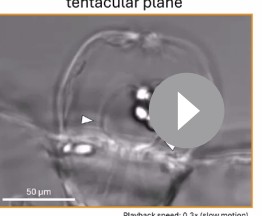
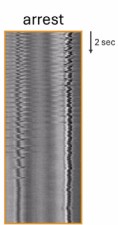

**Video 7.** High-speed imaging of balancer ciliary beating along with kymographs from two individual balancer ciliary clusters. The recording was acquired at 100 frames per second and played back at 30 frames per second (0.3×). The white line overlaid on the kymographs indicates the current time point shown in the video. This example captures a ciliary re-beat event in the tentacular plane.

https://elifesciences.org/articles/108420/figures#video7

**Video 6.** High-speed imaging of balancer ciliary beating along with kymographs from two individual balancer ciliary clusters. The recording was acquired at 100 frames per second and played back at 30 frames per second (0.3×). The white line overlaid on the kymographs indicates the current time point shown in the video. This example captures a ciliary arrest event in the tentacular plane.

https://elifesciences.org/articles/108420/figures#video6

when viewed along the tentacular plane. This morphology suggests a potential role for bridge neurons in modulating inter-quadrant coordination. By linking ANN activity across orthogonal planes, these neurons may contribute to the dynamic regulation of CBF patterns by facilitating the spread of activity across orthogonal planes, thus supporting coordinated beating throughout the aboral organ.

## Discussion
### An aboral nerve net for ciliary coordination

In this study, we generated the first high-resolution connectome of the ctenophore nervous system by reconstructing approximately 900 cells from the aboral organ of a 5-day-old *M. leidyi* cydippid by volumetric electron microscopy. Among the reconstructed cells, we identified a previously undescribed type of syncytial neuron—here referred to as aboral nerve net (ANN)—with multiple nuclei. The ANN has a morphology distinct from that of the SNN reported for 1-day-old cydippids (*Burkhardt et al., 2023*; *Sachkova et al., 2021*). The ANN neurons described here are morphologically and anatomically distinct from the body wall SNN. Instead of forming discrete nodal swellings or a 'beads-on-a-string' organization typical for the SNN, ANNs extend through intercellular spaces, weaving between neighbouring cells in a smooth, interdigitating fashion. The presence of four SNN-type anastomosing neurons more laterally in the aboral organ in our dataset (*Figure 2—figure supplement 1*) further underlines the distinct identity of ANNs. The neurotransmitter content and activation dynamics of these nerve-net neurons remain unresolved. However, the combination of circuit architecture and behavioural outputs provides the first insight into the function of nerve-net neurons in a ctenophore. A parallel vEM study of a

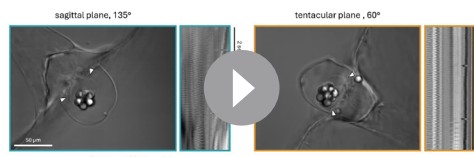

**Video 8.** High-speed imaging of balancer ciliary beating in the aboral organ tilted relative to gravity, illustrating gravity-induced asymmetry. The example on the left shows an aboral organ tilted by approximately 135° in the sagittal plane, while the example on the right shows one tilted by approximately 60° in the tentacular plane. For each case, the left and right balancer ciliary clusters are displayed separately. Ciliary beating is shown alongside kymographs extracted from two representative clusters. Recordings were acquired at 100 frames per second and are played back at 30 frames per second (0.3×). A white vertical line overlaid on each kymograph indicates the current frame shown in the video. These examples demonstrate asymmetrical balancer activity consistent with geotactic responses reported by *Tamm, 1980*.

https://elifesciences.org/articles/108420/figures#video8

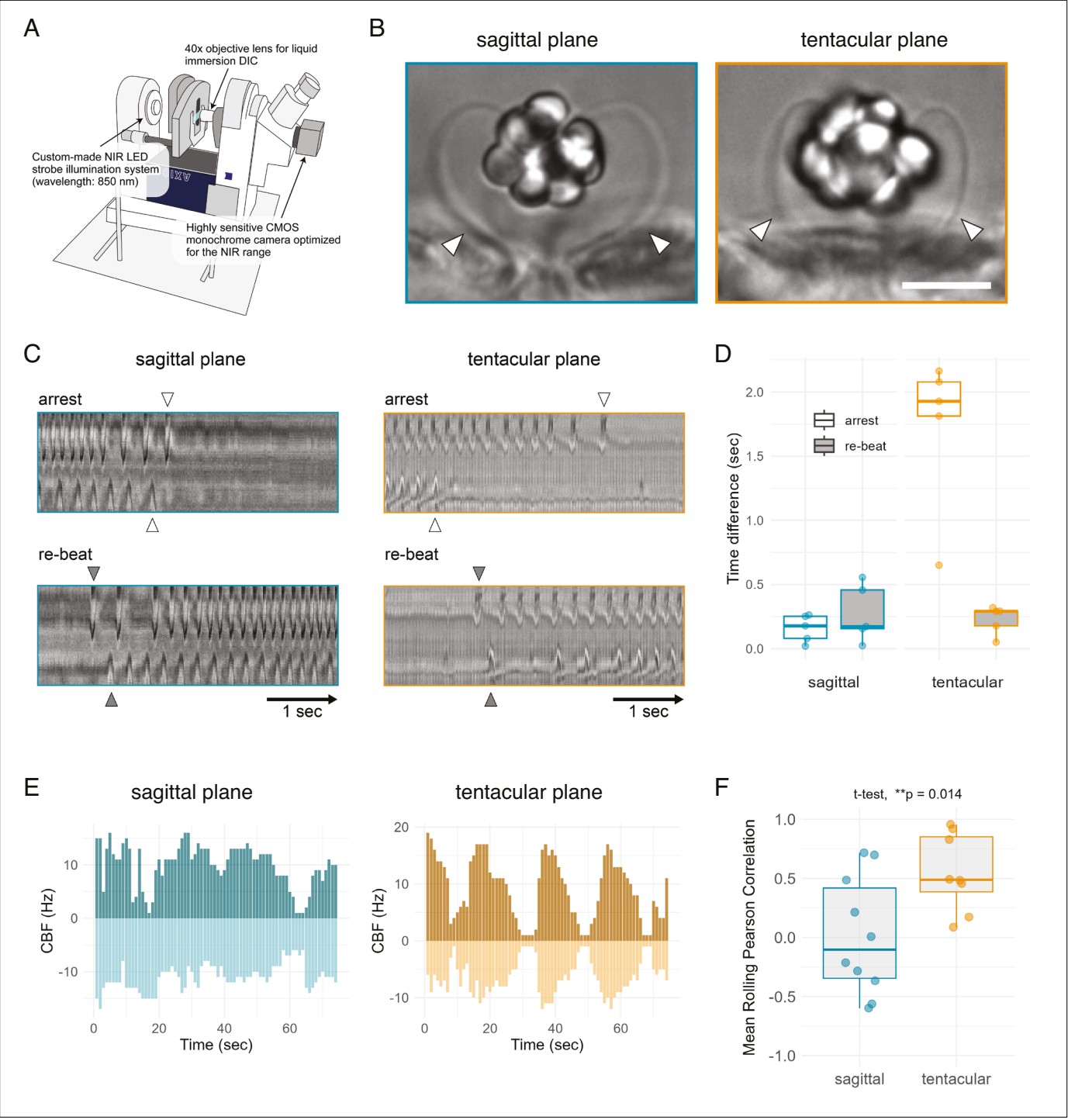

**Figure 4.** Analysis of ciliary beating, arrests and re-beat across balancers in the four quadrants. (**A**) Schematic diagram of the differential interference contrast (DIC) microscopy setup used to image the movement of balancer cilia. The microscope was tilted 90° so that the stage was positioned vertically. We used a 40× objective lens and a monochrome CMOS camera sensitive to near-infrared (NIR) light that was synchronized with an 850 nm strobe light source. (**B**) Representative DIC images of the aboral organ viewed along the sagittal (left) and tentacular (right) planes. Arrowheads indicate the balancer cilia selected for kymograph-based analysis in each orientation. Scale bar: 25 μm. (**C**) Representative kymographs showing arrest and re-beat events of balancer cilia. Left and right balancer cilia were simultaneously recorded in either the sagittal (left) or tentacular plane (right). Arrowheads mark the onset of arrest (open) and re-beat (filled); horizontal arrows represent 1-s intervals. (**D**) Boxplots showing time differences in the onset of arrest (white boxes) and re-beat (grey boxes) between left and right balancer cilia. Colours of the box outlines indicate the imaging plane: sagittal (blue) or tentacular (orange). Each dot represents a single larva. (**E**) Bar plots showing ciliary beat frequency (CBF) dynamics of left and right balancers (top and bottom, respectively) recorded from the sagittal (left) or the tentacular (right) planes. Positive and negative values represent opposite sides within

*Figure 4 continued on next page*

*Figure 4 continued*

each plane. (**F**) Mean rolling Pearson correlation of left–right balancer CBF calculated with a 20-frame window. Each dot represents an individual larva. Balancers on the tentacular plane showed significantly higher correlation than those on the sagittal plane (*t*-test, p = 0.014).

The online version of this article includes the following source data and figure supplement(s) for figure 4:

**Source data 1.** Time differences between arrest and re-beat responses of balancer cilia in sagittal and tentacular planes.

**Source data 2.** Ciliary beat frequency (CBF) values over time for left and right balancers in the sagittal plane.

**Source data 3.** Ciliary beat frequency (CBF) values over time for left and right balancers in the tentacular plane.

**Source data 4.** Rolling Pearson correlation values of CBF between left and right balancers in sagittal and tentacular planes.

**Figure supplement 1.** Ciliary beat frequency (CBF) dynamics of balancer cells in additional samples.

**Figure supplement 1—source data 1.** Ciliary beat frequency (CBF) time-series data for left and right balancers in the sagittal plane (recording 1).

**Figure supplement 1—source data 2.** Ciliary beat frequency (CBF) time-series data for left and right balancers in the sagittal plane (recording 2).

**Figure supplement 1—source data 3.** Ciliary beat frequency (CBF) time-series data for left and right balancers in the sagittal plane (recording 3).

**Figure supplement 1—source data 4.** Ciliary beat frequency (CBF) time-series data for left and right balancers in the tentacular plane (recording 1).

**Figure supplement 1—source data 5.** Ciliary beat frequency (CBF) time-series data for left and right balancers in the tentacular plane (recording 2).

**Figure supplement 1—source data 6.** Ciliary beat frequency (CBF) time-series data for left and right balancers in the tentacular plane (recording 3).

1-day-old cydippid describes a 'condensed' SNN in the aboral region, which displays a more compact morphology compared to body wall SNNs (*Ferraioli et al., 2025*). This condensed part of the SNN has a similar position and shape to the ANN we describe here. Furthermore, only the condensed part of the nerve net contained synapses. The main synaptic target of the condensed nerve net is bridge and balancer cells, which matches our observations. Therefore, the 5-day-old seems to have two morphologically and physically distinct neuronal cell types, while in the 1-day-old they seem to be not physically separated, even though they comprise two domains with different ultrastructure. It is unclear whether this is due to developmental differences or the difficulty of reliably reconstructing the neurites of the SNN.

ANNs form no synaptic output to classical effector organs such as muscles or comb plates. All observed output synapses were directed to balancer ciliary cells, bridge cells, or other sensory cells. This suggests that ANNs are unlikely to serve as motor drivers (premotor neurons or motoneurons) in the conventional sense. Rather, our findings suggest that the ANN network has a coordinating function to synchronize the activity of balancer cilia. This departs fundamentally from reflex-arc models of simple nervous systems (*Arendt, 2021*; *Jékely, 2011*) and instead supports the idea that coordination, not command, may have been a foundational role of ancestral neurons (*Keijzer et al., 2013*; *de Wiljes et al., 2015*). Such an organization may have been retained in modern ctenophores ever since they evolved a pelagic lifestyle and a balancer (*Stanley and Stürmer, 1983*).

How could the aboral nerve net exert a coordinating function? Our circuit reconstruction and high-speed imaging experiments suggest that the nerve net coordinates balancer ciliary beating, arrests and beat frequency (*Figure 5*). We observed that these changes in ciliary dynamics span multiple temporal scales from rapid arrests and re-beats to slower drifts in frequency, indicating hierarchical tuning of activity across balancers, implying higher-level coordination. We found differences in the innervation patterns of ANNs across the sagittal and tentacular planes. The two smaller ANNs targeted a pair of balancers (either Q1–Q2 or Q3–Q4) across the sagittal plane. This innervation pattern corresponded with plane-specific differences in the dynamics of balancer cilia. Across the sagittal but not the tentacular plane, the closures of balancer cilia were near-simultaneous, within less than one beat phase. Once all cilia have arrested, the re-beats occur nearly synchronously (within less than one beat phase) across all four quadrants, implying a global synchronizer. Since the large ANN innervates balancers in all four quadrants, this neuron could account for the near-simultaneous re-beat of the balancers.

The observation that balancer cilia beat spontaneously, even in the absence of external tilt, suggests that they are active sensory oscillators rather than static stretch sensors. Their spontaneous beating could set a dynamic baseline of sensitivity, which can then be modulated by ANN inputs or sensory changes during tilt. Such a dynamic system may be more sensitive to small deflections and be more responsive (*Lowe, 1997*). Thus, the regulated beating of balancer cilia should not be seen as noise, but as an adaptive feature that enables flexible and robust graviceptive responses. The ctenophore

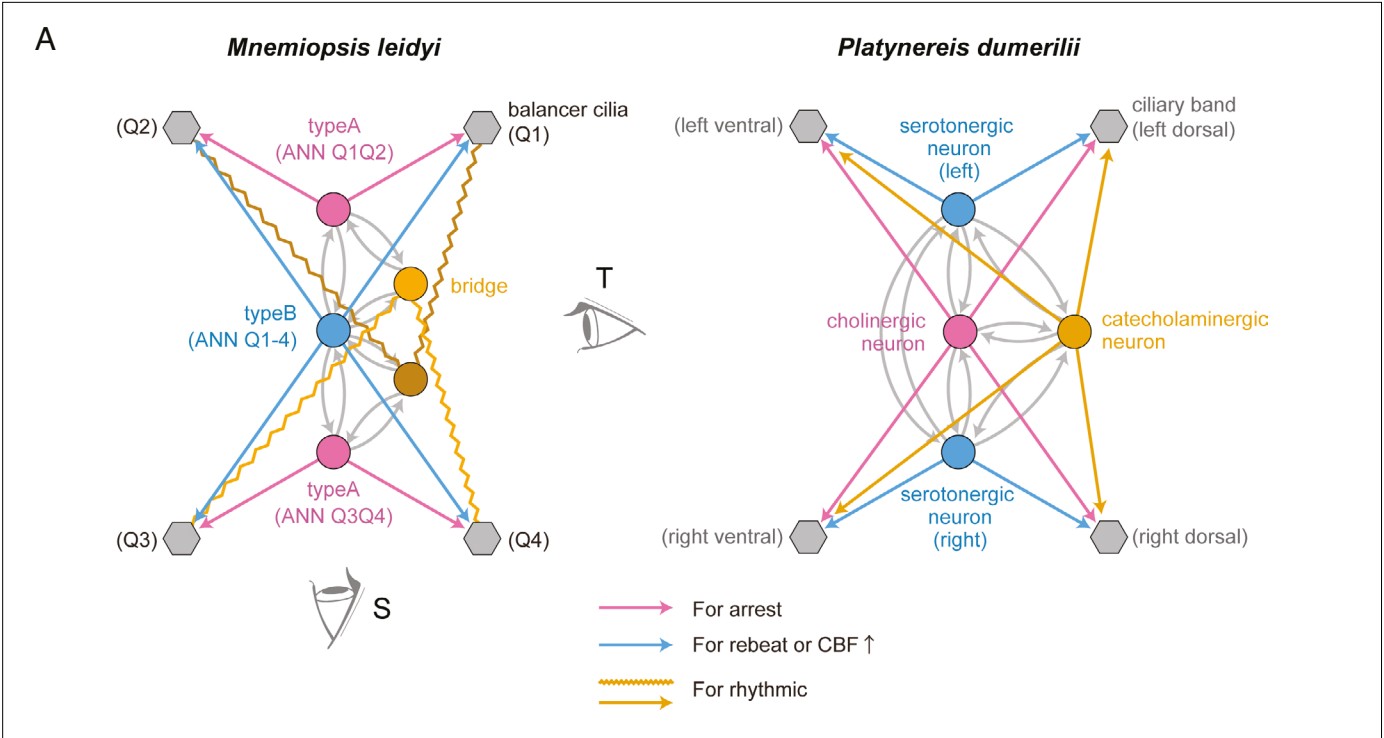

**Figure 5.** Comparison of neural circuits regulating ciliary movement. (**A**) Neurons are represented by circles, colour-coded to indicate analogous functions in ciliary control. Ciliated cells are shown in grey. Synapses are indicated by arrows, with magenta representing synapses that induce ciliary arrest and blue representing synapses that induce ciliary re-beat or an increase in ciliary beating frequency. (Left) Neural circuit of the *M. leidyi* gravi-sensory organ. Bridge cells (yellow squares) are suggested to be electrically coupled (indicated by yellow zigzag lines), implying a potential involvement in feedback mechanisms between neurons and ciliated cells. (Right) Neural circuit regulating the beating of cilia in the prototroch ciliary band of larval *P. dumerilii*. Serotonergic neurons (Ser-h1 neurons, blue) activate ciliary movement, while a cholinergic neuron (MC neuron, magenta) induces ciliary arrest.

balancer may thus use active ciliary oscillations for enhanced sensorimotor integration similar to other sensory systems (*Wan, 2023*).

How could ANN neurons regulate this active oscillatory balancer system? Coordinated ANN inputs may regularly reset responsiveness by arresting cilia and modulate sensitivity by inducing periodic frequency fluctuations. This synaptic regulation may ensure uniform sensitivity to sensory load and the integration of potential modulatory inputs. Ciliary coordination could be further enhanced locally in each balancer by electrical coupling via gap junctions. Although unresolved in our EM, the expression of innexin genes indicates their presence within balancer clusters (*Ortiz et al., 2023*).

An alternative model for the function of the nerve net would be a feedforward sensory–motor system, in which balancer cells provide mechanosensory input to motor effectors via the nerve net, similar to a reflex arc (*Figure 5*). None of our observations support such a sensory–motor model. There are no synaptic pathways from balancer cells or any other sensory cells to the nerve net. The only synaptic input to ANNs comes from the bridge cells (discussed below) and from each other. The three synaptically interconnected ANNs may generate endogenous rhythm that controls balancer cilia and is influenced by bridge input. ANNs may also be influenced by neuropeptides secreted by other aboral organ neurons. Such chemical inputs may underlie the flexibility of gravitaxis and its modulation by other cues (e.g. light). Overall, the coordination model parsimoniously explains both the ANN wiring topology and the observed dynamics, whereas a simple feedforward reflex does not.

The functional organization of the ANN circuit is reminiscent of the ciliomotor circuit in the larvae of the marine annelid *Platynereis dumerilii* (*Figure 5*). Here, the activation of cholinergic motor neurons leads to the coordinated arrest of cilia on all multiciliary bands across body segments (*Verasztó et al., 2017*). This is antagonized by serotonergic midline-crossing neurons that drive ciliary re-beat (*Calderón et al., 2024*; *Verasztó et al., 2017*). The ctenophore balancer-circuit is the second synaptically mapped ciliomotor circuit that convergently evolved analogous ciliary control mechanisms to the annelid and other cilimotor systems (*Marinković et al., 2020*; *Moroz, 2015*; *Roberts et al., 2022*).

## Bridge cells as feedback regulators of ciliary rhythms

In our volumetric reconstruction, bridge cells were identified as the only cell type providing synaptic input to the ANN network, positioning them as key upstream regulators of the ciliary coordination system in the aboral organ. Unexpectedly, no direct afferent connections from the balancers to the ANN were found; all balancer cells act as 'sink nodes' (inputs only; *Figure 3—figure supplement 1*). Thus, bridge cells remain the sole neuronal input to the ANN. Originally described morphologically by *Tamm and Tamm, 2002*, bridge cells extend thin projections towards the base of the balancer cilia (*Tamm, 2014a*; *Tamm and Tamm, 2002*). Based on this spatial relationship, they may receive information about the physical state of the cilia—such as mechanical load or phase—or possibly local field potential changes (*Jurisch-Yaksi et al., 2024*; *Sheu et al., 2022*). These features suggest that bridge cells may act as transducers, converting local ciliary signals into neural inputs directed towards the ANN circuit.

As revealed by our high-speed imaging, CBF exhibited rhythmic changes over time that differed between anatomical planes (*Figure 4E, F*), indicating spatially distinct coordination dynamics. These CBF rhythms are likely generated by the ANN circuit, while bridge cells may function to stabilize or adjust this rhythmic output in accordance with the real-time state of the cilia. This way, bridge cells may provide feedback about the state of balancer cilia to the coordination circuit, serving as a core element of a self-regulating rhythm-control system.

This architecture can be interpreted as a recurrent information loop, in which output from the ANN circuit modulates ciliary activity, and the resulting ciliary state is in turn relayed back to the ANN via bridge cells. Such a structure suggests a loop-based coordination mechanism (*Arshavsky, 2003*; *Selverston, 2010*), where the central circuit adapts its timing based on feedback from its target. Although the electrophysiological properties of bridge cells remain to be elucidated, their recursive integration into the ANN circuit indicates that they may function in maintaining the robustness and flexibility of endogenous ciliary rhythms (*Kennedy and Weissbourd, 2024*).

## Future directions

Our connectomic reconstruction of the ctenophore aboral organ provides a first framework for understanding how the ANN neurons may coordinate the activity of balancer cilia. Establishing causality, however, will require new experimental approaches. For example, a direct test of the neuronal control of balancer cilia could involve the targeted ablation of ANN neurons combined with ciliary imaging. Alternatively, calcium imaging of ANN activity while imaging cilia could indicate a physiological link between neural signalling and ciliary movement. Such experiments will require advances in genetic delivery methods and reporter constructs in ctenophores. We are actively developing microinjection protocols for mRNA and DNA delivery into zygotes to enable these future applications.

Future experiments could also explore how orientation affects the response of balancer cilia. For example, when the statolith is suspended below the cilia (the 'balancer-down' posture), ciliary beating patterns may differ from what we observed here in the 'balancer-up' configuration. Orientation-dependent analyses, together with multi-sensory paradigms (e.g. light or other stimuli), will help determine how the balancer combines multiple inputs to generate motor output and how the ANN shapes its dynamics. Finally, further work is needed to address the molecular composition of ctenophore neurons, including their neurotransmitters, ion channels, and modulators, and their functions in ANN signalling and ciliary coordination.

## Conclusion

How do differences in balancer-cilia coordination across anatomical planes shape behaviour? In *Mnemiopsis*, the animal swims upright with its mouth facing upward and performs 90° rotational turns when hunting prey or responding to disturbances (*Colin et al., 2010*; *Courtney et al., 2020*; *Waggett, 1999*). Large rotational manoeuvres may involve balancer cells in the tentacular plane, while sagittal-plane balancers may contribute to postural stabilization (*Tamm, 2015*).

Our findings reveal a circuit composed of ANN neurons and bridge cells that coordinates ciliary beating rhythms in space and time without direct sensory input and features a recursive architecture in which ciliary outputs may feed back to the circuit via bridge cells. Such a configuration suggests that even with a limited number of cells, the system is capable of flexible and robust rhythm control.

As *Jager et al., 2011* once described, this organ evokes the metaphor of a 'ciliary brain' (*Jager et al., 2011*). The network we describe here orchestrates output through dynamic, self-regulating coordination. Future work could uncover these dynamics, visualized through live imaging of the entire cell ensemble, to offer a window into the core functions of a unique nervous system.

## Materials and methods

### Specimen preparation, vEM, and image processing

Larvae of *M. leidyi* 5-day-old (5 days post-hatching) were cryofixed using a high-pressure freezing apparatus (HPM Live μ, CryoCapCell) and immediately transferred to liquid nitrogen for storage. The frozen samples were processed in a substitution medium containing 2% (wt/vol) osmium tetroxide and 0.5% uranyl acetate in acetone, using a cryo-substitution device (EM AFS-2, Leica).

Cryo-substitution was carried out by gradually raising the temperature from –90 to –70°C over 4 hr, then returning to –20°C over 2 hr, and finally to room temperature over 2 hr. The samples were then embedded in epoxy resin (EMbed 812, Electron Microscopy Sciences).

Serial ultrathin sections of 50 nm thickness were prepared using a Leica UC7 ultramicrotome and a 45° DiATOME diamond knife. Conductive indium tin oxide-coated glass slides (ITO Glass, UQG Optics) were treated with air glow discharge using the PELCO easiGlow system (Ted Pella, Inc) to enhance section adhesion and improve hydrophilicity. This process rendered the glass surface negatively charged.

Section ribbons were collected on prepared glass slides, gently dried to ensure proper stretching, and firmly adhered to the glass surface. The sections were stained with UranyLess and lead citrate (Reynolds) using airless staining procedures. The glass slides were mounted on an STEM-specific stage (Zeiss) using Copper Foil EMI Shielding Tape (3M).

Imaging was done on a Gemini SEM 500 (Zeiss), equipped with SmartSEM and Atlas 5 imaging software (Zeiss), using the Inlens detector. A total of 620 serial sections, each 50 nm thick, were analysed. The imaging resolution was 2.8 nm/pixel, and the acceleration voltage was set to 1.5 kV. The imaging time for each tile was 54 min and 21 s, with a pixel size of 2.8 nm, a tile size of 91.8 × 91.8 μm (32,768 × 32,768 pixels), and a dwell time of 3.0 μs.

### Image-stack alignment and export for CATMAID

To process the image stack, we utilized the TrakEM2 plugin of FIJI (ImageJ) (version 2.0.0-rc-15/1.49k/Java 1.6.0_24 (64-bit)—2014). A project was created, and all TIFF images were imported using the 'import sequence as grid' function. Subsequently, the following filters were applied sequentially: Invert, Equalize Histogram, and Gaussian Blur.

The alignment process consisted of three stages, each progressively refining the spatial accuracy (rigid, affine, and elastic).

Initially, a rigid alignment was run with the following parameters: least squares mode (linear feature correspondences), encompassing the entire layer range with the first layer as the reference. Only visible images were used, without propagation. The alignment was executed with an initial Gaussian blur of 1.6 pixels, three steps per scale octave, a minimum image size of 512 pixels, and a maximum of 2048 pixels. Additional parameters included a feature descriptor size of 8, orientation bins of 8, and a closest ratio of 0.92. The alignment allowed clearing the cache, using 32 feature-extraction threads, a maximal alignment error of 100 pixels, a minimal inlier ratio of 0.20, and a minimum of 12 inliers. The expected and desired transformations were set to rigid, with testing multiple hypotheses (tolerance: 5.00 pixels) and considering up to five neighbouring layers, giving up after five failures. Regularization was done with a maximal iteration of 1000, a maximal plateau width of 200, and a rigid lambda of 0.10.

Next, we applied an affine alignment step with similar parameters, except the expected and desired transformations were set to affine. The minimal image size was reduced to 64 pixels, while the other parameters (Gaussian blur, feature descriptor size, inliers, and testing hypotheses) remained unchanged to ensure consistent processing.

Finally, we ran two iterations of elastic alignment to fine-tune the spatial data. Key parameters included a block-matching layer scale of 0.05, a search radius of 200 pixels, a block radius of 2000 pixels, and a resolution of 60. Correlation filters were employed with a minimal PMCC r of 0.10, a

maximal curvature ratio of 1000, and a maximal second-best r/best r of 0.90. A local smoothness filter was applied with the approximate local transformation set to affine, a local region sigma of 1000 pixels, and an absolute maximal local displacement of 10 pixels (relative maximal displacement: 3.00). Pre-aligned layers were tested for up to four neighbouring layers. The elastic alignment used a rigid approximation, maximal iterations of 3000, a plateau width of 200, spring mesh stiffness of 0.01, and a maximal stretch of 2000 pixels. A legacy optimizer was employed to enhance performance.

After each alignment stage, the project was saved as an XML file under a unique name to preserve iterative progress. Finally, the images were exported from FIJI using TrakEM2 in a format compatible with CATMAID.

## Neuron tracing, synapse annotation, and review

For skeletonization, annotation and tagging, we used CATMAID installed on a local server (*Saalfeld et al., 2009*; *Schneider-Mizell et al., 2016*). To mark the locations of cell bodies, we placed tags at the approximate centre of each nucleus within the dataset. At each nuclear centre, the radius of the single node was adjusted according to the size of the cell body in that specific layer. All skeletons were rooted at their respective cell bodies, and the root nodes were tagged as 'soma'. Synapses were identified based on four key structural features: the cell membrane, synaptic vesicles, the endoplasmic reticulum, and mitochondria. Most synapses could be verified across consecutive sections, ensuring accurate annotation and connectivity mapping.

## Cell-type nomenclature, annotations, and connectome analysis

We assigned each cell a specific cell-type name based on its morphological or functional category (e.g. balancer, bridge, large granular, ciliated grooves, dense vesicle cell, dome, epithelial floor (ef), intracellular multiciliated cells (imc), lamellate bodies (lb), lithocyte, plumose cell, and ANN), resulting in a total of 12 primary cell types.

In addition, some cells lacked distinctive features and were instead categorized into one of four general ciliated types, based on the presence and number of cilia: biciliated (biC), monociliated (monoC), multiciliated (multiC), or non-ciliated (non-C).

To indicate spatial positioning, we appended the quadrant number to the cell-type name using an underscore. For instance, cells located in the first quadrant were labelled '_Q1', while those situated between two quadrants were labelled with both (e.g. 'Q1Q2').

We further distinguished cells of the same type within the same region by serial numbering. This way, each cell in the volume has a unique name string. Each cell was also assigned multiple annotations, which can be utilized to query the database via CATMAID or the CATMAID API (e.g. using the R catmaid package; *Bates et al., 2020*). These annotations provide a structured and precise framework for identifying and analysing specific cells, facilitating robust data integration and retrieval from the dataset.

## Imaging the activity of balancer cilia

For ciliary imaging, we used the cydippid stage of *M. leidyi* at 5-day-old. A coverslip with a thin layer of Vaseline applied to its two edges was gently placed over a glass slide. Filtered natural seawater and cydippids were placed beneath the coverslip. The orientation of the larvae was adjusted by carefully sliding the coverslip, and the larvae were gently immobilized by applying slight pressure.

To ensure consistency in gravitational input, we selected only larvae whose aboral–oral axis was oriented nearly parallel to the gravitational vector (within 20° tilt). This minimized the potential influence of larval tilt on left–right differences in balancer cilia activity, as all balancers received comparable stimulation from the statolith.

This orientation corresponds to a 'balancer-up' orientation, in which the aboral organ faces upward. We selected this configuration not only to ensure consistent stimulation from the statolith across individuals, but also because it facilitates intuitive interpretation of balancer function—as if balancing a stone on top. Moreover, this orientation matches the orientation used in electron microscopy and was technically advantageous for high-speed imaging under vertical-stage alignment. Using a consistent orientation throughout minimized potential confusion when comparing ultrastructural and functional data.

Because the tilt angle strongly affects the beat frequency of balancer cilia via asymmetric mechanical loading by the statolith (*Tamm, 1980*), our quantitative analysis was restricted to recordings that met the above orientation criterion (tilt <20°). In this orientation, gravitational input from the statolith is approximately symmetrical across the left and right balancers, allowing meaningful comparisons of their activity. To illustrate how excessive tilt induces asymmetric stimulation, *Video 8* presents two examples of larvae outside the permitted range: one tilted by ~135° in the sagittal plane and the other by ~60° in the tentacular plane. In both cases, strong asymmetries in balancer ciliary beating are evident, consistent with previous observations of gravity-induced geotactic responses (*Tamm, 1980*).

Balancer-cilia movement was imaged using a differential interference contrast (DIC) microscope (Zeiss Axio Imager.M2) equipped with a 40× glycerine-immersion objective (LD LCI Plan-Apochromat 40×/1.2 Imm Corr DIC M27). Images were acquired with a high-sensitivity monochrome CMOS camera (UI-3360CP-NIR-GL Rev.2, iDS), optimized for near-infrared detection. Illumination was provided by a custom-built 850 nm LED strobe system, synchronized with camera exposure using BohNavi software. Recordings were acquired at 640 × 480 pixels, 100 frames per second (fps), for a duration of 2 min, using 0.05-ms pulse illumination from the LED.

For beat frequency analysis, kymographs were generated using the Multi Kymograph plugin in Fiji. One-second segments were manually extracted from the kymographs, and CBFs were visually counted for each manually defined ROI. These manually determined frequencies were then used for subsequent computational analysis.

Traces of left and right cilia were extracted, cleaned to remove missing or non-finite values, and indexed by frame. Data were saved as RDS objects for downstream analysis. Mean values were plotted as bar graphs with left and right cilia plotted as positive and negative bars, respectively, to visually distinguish their activity.

To assess coordination between left and right cilia, we computed rolling Pearson correlation coefficients over a window of 20 frames. For each dataset, we excluded windows with insufficient valid data or no variation. Time series of correlation coefficients were plotted and summarized as mean values. These mean correlations were then compared across planes (sagittal vs. tentacular) using Welch's *t*-test. The resulting statistics were visualized as boxplots with jittered data points.

## Acknowledgements

This work was supported by the Japan Society for the Promotion of Science (JSPS) Overseas Research Fellowships, the Grass Foundation, the Kavli Foundation, and the Marine Biological Laboratory (MBL). This project also received funding from the European Research Council (ERC) under the European Union's Horizon 2020 research and innovation programme (grant agreement No. 101020792). We thank Dr Réza Shahidi for sectioning and imaging, Paulina Cherek for high-pressure freezing and sample preparation, and Iva Verbanac for help with tracing. We are also grateful to Dr Chris Bjornsson, the MBL Central Microscopy Facility, Dr Shoji A Baba, and Dr Kogiku Shiba for their assistance with microscopy. We also thank Dr Sidney L Tamm and Dr Steven HD. Haddock for their valuable feedback on the manuscript.

## Additional information

### Competing interests

Gáspár Jékely: Reviewing editor, eLife. The other authors declare that no competing interests exist.

### Funding

| Funder | Grant reference number | Author |
| --- | --- | --- |
| European Research Council | 101020792 | Gáspár Jékely |
| Grass Foundation | | Kei Jokura |

| Funder | Grant reference number | Author |
|---|---|---|
| The Japan Society for the Promotion of Science (JSPS) | 1200332 | Kei Jokura |
| The Kavli Foundation | | Kei Jokura |

The funders had no role in study design, data collection, and interpretation, or the decision to submit the work for publication.

### Author contributions
Kei Jokura, Conceptualization, Data curation, Software, Formal analysis, Funding acquisition, Validation, Investigation, Visualization, Methodology, Project administration, Writing – review and editing, Writing – original draft; Sanja Jasek, Data curation, Software, Formal analysis, Visualization, Methodology, Writing – review and editing; Lara Niederhaus, Formal analysis, Investigation, Visualization; Pawel Burkhardt, Methodology, Ctenophore culture and provision of animals for vEM; Gáspár Jékely, Conceptualization, Data curation, Software, Formal analysis, Supervision, Funding acquisition, Writing – original draft, Project administration, Writing – review and editing

### Author ORCIDs
Kei Jokura ⓘ https://orcid.org/0000-0002-5347-0351
Sanja Jasek ⓘ https://orcid.org/0000-0001-6844-4319
Pawel Burkhardt ⓘ https://orcid.org/0000-0001-9826-057X
Gáspár Jékely ⓘ https://orcid.org/0000-0001-8496-9836

Reviewer #1 (Public review): https://doi.org/10.7554/eLife.108420.3.sa1
Reviewer #2 (Public review): https://doi.org/10.7554/eLife.108420.3.sa2
Reviewer #3 (Public review): https://doi.org/10.7554/eLife.108420.3.sa3
Author response https://doi.org/10.7554/eLife.108420.3.sa4

## Additional files

### Supplementary files
MDAR checklist

### Data availability
The EM image stacks, including all traces and annotations, can be viewed at https://catmaid.jekelylab.ex.ac.uk or https://www.w3id.org/jekelylab-catmaid. The dataset encompasses all EM images (in JPG format), skeletons, meshes, node tags, connectors, and annotations. The raw images, TrakEM2 alignment file, catmaid tiles, and reconstruction have also been deposited in the Electron Microscopy Public Image Archive (EMPIAR) with the accession code EMPIAR-13030. Additionally, we provide all R scripts used for data acquisition and figure generation (*Jokura et al., 2025*). All plots, figures (including anatomical renderings), and figure layouts can be fully reproduced using the provided R scripts. While the scripts are mostly organized by figure, some general-purpose scripts—for tasks such as loading libraries, accessing CATMAID data, and defining common functions—are shared across multiple figures.

The following dataset was generated:

| Author(s) | Year | Dataset title | Dataset URL | Database and Identifier |
|---|---|---|---|---|
| Jokura K, Jasek S, Niederhaus L, Burkhardt P, Jékely G | 2025 | Volume EM stack of the ctenophore aboral organ | https://www.ebi.ac.uk/empiar/EMPIAR-13030/ | EMPIAR, EMPIAR-13030 |

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
