## [Editor Report · eLife Assessment]

This **fundamental** work significantly advances our understanding of gravity sensing and orientation behavior in the ctenophore, an animal of major importance in understanding the evolution of nervous systems. Through comprehensive reconstruction with volumetric electron microscopy, and time-lapse imaging of cilia motion, the authors provide **compelling** evidence that the aboral nerve net coordinates the activity of balancer cilia. The resemblance to the ciliomotor circuit in marine annelids provides a fascinating example of how neural circuits may convergently evolve to solve common sensorimotor challenges.

---

## [Referee Report · Reviewer #1 (Public review)]

Summary:

This work presents an interesting circuit dissection of the neural system allowing a ctenophore to keep its balance and orientation in its aquatic environment by using a fascinating structure called the statocyst. By combining serial-section electron microscopy with behavioral recordings, the authors found a population of neurons which exists as a syncytium and could associate these neurons with specific functions related to controlling the beating of cilia located in the statocyst. The type A ANN neurons participate in arresting cilia beating, and the type B ANN neurons participate in resuming cilia beating and increasing their beating frequency.

Moreover, the authors found that bridge cells are connected with the ANN neurons, giving them the role of rhythmic modulators.

From these observations, the authors conclude that the control is coordination instead of feedforward sensory-motor function, a hypothesis that had been put forth in the past but could not be validated until now. They also compare it to the circuitry implementing a similar behavior in a species that belongs to a different phylum where the nervous system is thought to have evolved separately.

Therefore, this work significantly advances our knowledge of the circuitry implementing the control of the cilia that participate in statocyst function which ultimately allow the animal to correct its orientation. It explains how the nervous system allows an animal to solve a specific problem and puts it in an evolutionary perspective showing a convincing case of convergent evolution.

Strengths:

The evidence for how the circuitry is connected is convincing. Pictures of synapses showing the direction of connectivity are clear and there are good reasons to believe that the diagram inferred is valid, even though we can always expect that some connections are missing.

The evidence for how the cilia change their beating frequency is also convincing, and the paradigm and recording methods seem pretty robust.

The authors achieved their aims and the results support their conclusions. This work impacts its field by presenting a mechanism by which ctenophores correct their balance, which will provide a template for comparison with other sensory systems.

---

## [Referee Report · Reviewer #2 (Public review)]

Summary:

In this manuscript, the authors describe the production of a high-resolution connectome for the statocyst of a ctenophore nervous system. This study is of particular interest because of the apparent independent evolution of the ctenophore nervous system. The statocyst is a component of the aboral organ, which is used by ctenophores to sense gravity and regulate the activity of the organ's balancer cilia. The EM reconstruction of the aboral organ was carried out on a five-day old larva of the model ctenophore Mnemiopsis leidyi. To place their connectome data in a functional context, the authors used high-speed imaging of ciliary beating in immobilized larvae. With these data, the authors were able to model the circuitry used for gravity sensing in a ctenophore larva.

Strengths:

Because of it apparently being the sister phylum to all other metazoans, Ctenophora is a particularly important group for studies of metazoan evolution. Thus, this work has much to tell us about how animals evolved. Added to that is the apparent independent evolution of the ctenophore nervous system. This study provides the first high-resolution connectomic analysis of a portion of a ctenophore nervous system, extending previous studies of the ctenophore nervous system carried out by Sid Tamm. As such it establishes the methodology for high-resolution analysis of the ctenophore nervous system. While the generation of a connectome is in and of itself an important accomplishment, the coupling of the connectome data with analysis of the beating frequency of balancer cell cilia provides a functional context for understanding how the organization of the neural circuitry in the aboral organ carries out gravity sensing. In addition, the authors identified a new type of syncytial neuron in Mnemiopsis. Interestingly, the authors show that the neural circuitry controlling cilia beating in Mnemiopsis shares features with the circuitry that controls ciliary movement in the annelid Platynereis, suggesting convergent evolution of this circuity in the two organisms. The data in this paper are of high quality, and the analyses have been thoroughly and carefully done.

Weaknesses:

The paper has no obvious weaknesses.

Comments on revisions:

The authors have satisfactorily addressed the minor issues that I brought up in my original review.

---

## [Referee Report · Reviewer #3 (Public review)]

Summary:

It has been a long time since I enjoyed reviewing a paper as much as this one. In it, the authors generate an unprecedented view of the aboral organ of a 5-day old ctenophore. They proceed to derive numerous insights by reconstructing the populations and connections of cell types, with up to 150 connections from the main Q1-4 neuron.

Strengths:

The strengths of the analysis are the sophisticated imaging methods used, the labor-intensive reconstruction of individual neurons and organelles, and especially the mapping of synapses. The synaptic connections to and from the main coordinating neurons allow the authors to created a polarized network diagram for these components of the aboral organ. These connections give insight about the potential functions of the major neurons, which also giving some unexpected results, particularly the lack of connections from the balancer system to the coordinating system.

Weaknesses:

There were no significant weaknesses in the paper - only a slate of interesting unanswered questions to motivate future studies.

Comments on revisions:

This manuscript was already strong from the start, and I am fully satisfied with the revisions, which corrected a few glitches and points of clarification.

---

## [Author Response]

The following is the authors’ response to the original reviews.

**Public Reviews:**

**Reviewer #1 (Public review):**
Summary:This work presents an interesting circuit dissection of the neural system allowing a ctenophore to keep its balance and orientation in its aquatic environment by using a fascinating structure called the statocyst. By combining serial-section electron microscopy with behavioral recordings, the authors found a population of neurons that exists as a syncytium and could associate these neurons with specific functions related to controlling the beating of cilia located in the statocyst. The type A ANN neurons participate in arresting cilia beating, and the type B ANN neurons participate in resuming cilia beating and increasing their beating frequency.Moreover, the authors found that bridge cells are connected with the ANN neurons, giving them the role of rhythmic modulators.From these observations, the authors conclude that the control is coordination instead of feedforward sensory-motor function, a hypothesis that had been put forth in the past but could not be validated until now. They also compare it to the circuitry implementing a similar behavior in a species that belongs to a different phylum, where the nervous system is thought to have evolved separately.Therefore, this work significantly advances our knowledge of the circuitry implementing the control of the cilia that participate in statocyst function, which ultimately allows the animal to correct its orientation. It represents an example of systems neuroscience explaining how the nervous system allows an animal to solve a specific problem and puts it in an evolutionary perspective, showing a convincing case of convergent evolution.Strengths:The evidence for how the circuitry is connected is convincing. Pictures of synapses showing the direction of connectivity are clear, and there are good reasons to believe that the diagram inferred is valid, even though we can always expect that some connections are missing.The evidence for how the cilia change their beating frequency is also convincing, and the paradigm and recording methods seem pretty robust.The authors achieved their aims, and the results support their conclusions. This work impacts its field by presenting a mechanism by which ctenophores correct their balance, which will provide a template for comparison with other sensory systems.

Thank you very much for these comments.

Weaknesses:The evidence supporting the claim that the neural circuitry presented here controls the cilia beating is more correlational because it only relies on the fact that the location of the two types of ANN neurons coincides with the quadrants that are affected in the behavioral recordings. Discussing ways by which causality could be established might be helpful.

We have now added additional discussions in a new “Future Directions” section explaining that for example calcium imaging or targeted neuron ablations could be used in future work to establish causality. This would require the development of genetic delivery techniques to e.g. introduce GCaMP calcium sensor or transgenic reporters.

The explanation of the relevance of this work could be improved. The conclusion that the work hints at coordination instead of feedforward sensory-motor control is explained over only a few lines. The authors could provide a more detailed explanation of how the two models compete (coordination vs feedforward sensory-motor control), and why choosing one option over the other could provide advantages in this context.

We added a more detailed explanation about the two types of model and why we believe that a coordination model is more compatible with our connectome data.

“An alternative model for the function of the nerve net would be a feedforward sensory-motor system, in which balancer cells provide mechanosensory input to motor effectors via the nerve net, similar to a reflex arc. None of our observations support such a sensory-motor model. There are no synaptic pathways from balancer cells or any other sensory cells to the nerve net. The only synaptic input to ANNs comes from the bridge cells (discussed below) and from each other. The three synaptically interconnected ANNs may generate endogenous rhythm that controls balancer cilia and is influenced by bridge input. ANNs may also be influenced by neuropeptides secreted by other aboral organ neurons. Such chemical inputs may underlie the flexibility of gravitaxis and its modulation by other cues (e.g. light). Overall, the coordination model parsimoniously explains both the ANN wiring topology and the observed dynamics, whereas a simple feedforward reflex does not.”

Since the fact that the ANN neurons form a syncytium is an important finding of this study, it would be useful to have additional illustrations of it. For instance, pictures showing anastomosing membranes could typically be added in Figure 2.

We have now included a movie (Video 3) showing a volumetric reconstruction of a segment of an ANN neuron, which highlights the anastomosing morphology in greater detail than static images.

“Video 3. Volumetric reconstruction of a single ANN Q1-4 neuron showing syncytial soma (cyan) and nuclei (magenta). The rotating view highlights the anastomosing morphology, although not all fine details could be reconstructed due to data limitations.”

Also, to better establish the importance of the study, it could be useful to explain why the balancers’ cilia spontaneously beat in the first place (instead of being static and just acting as stretch sensors).

We have discussed in more detail why it may be important for the balancer cilia to beat.

“The observation that balancer cilia beat spontaneously, even in the absence of external tilt, suggests that they are active sensory oscillators rather than static stretch sensors. Their spontaneous beating could set a dynamic baseline of sensitivity, which can then be modulated by ANN inputs or sensory changes during tilt. Such a dynamic system may be more sensitive to small deflections and be more responsive [@Lowe1997]. Thus, the regulated beating of balancer cilia should not be seen as noise, but as an adaptive feature that enables flexible and robust graviceptive responses. The ctenophore balancer may thus use active ciliary oscillations for enhanced sensorimotor integration similar to other sensory systems [@Wan_2023].”

**Reviewer #2 (Public review):**
Summary:In this manuscript, the authors describe the production of a high-resolution connectome for the statocyst of a ctenophore nervous system. This study is of particular interest because of the apparent independent evolution of the ctenophore nervous system. The statocyst is a component of the aboral organ, which is used by ctenophores to sense gravity and regulate the activity of the organ’s balancer cilia. The EM reconstruction of the aboral organ was carried out on a five-day-old larva of the model ctenophore Mnemiopsis leidyi. To place their connectome data in a functional context, the authors used high-speed imaging of ciliary beating in immobilized larvae. With these data, the authors were able to model the circuitry used for gravity sensing in a ctenophore larva.Strengths:Because of it apparently being the sister phylum to all other metazoans, Ctenophora is a particularly important group for studies of metazoan evolution. Thus, this work has much to tell us about how animals evolved. Added to that is the apparent independent evolution of the ctenophore nervous system. This study provides the first high-resolution connectomic analysis of a portion of a ctenophore nervous system, extending previous studies of the ctenophore nervous system carried out by Sid Tamm. As such, it establishes the methodology for high-resolution analysis of the ctenophore nervous system. While the generation of a connectome is in and of itself an important accomplishment, the coupling of the connectome data with analysis of the beating frequency of balancer cell cilia provides a functional context for understanding how the organization of the neural circuitry in the aboral organ carries out gravity sensing. In addition, the authors identified a new type of syncytial neuron in Mnemiopsis. Interestingly, the authors show that the neural circuitry controlling cilia beating in Mnemiopsis shares features with the circuitry that controls ciliary movement in the annelid Platynereis, suggesting convergent evolution of this circuitry in the two organisms. The data in this paper are of high quality, and the analyses have been thoroughly and carefully done.Weaknesses:The paper has no obvious weaknesses.

We thank the reviewer for these comments.

**Reviewer #3 (Public review):**
Summary:It has been a long time since I enjoyed reviewing a paper as much as this one. In it, the authors generate an unprecedented view of the aboral organ of a 5-day-old ctenophore. They proceed to derive numerous insights by reconstructing the populations and connections of cell types, with up to 150 connections from the main Q1-4 neuron.Strengths:The strengths of the analysis are the sophisticated imaging methods used, the labor-intensive reconstruction of individual neurons and organelles, and especially the mapping of synapses. The synaptic connections to and from the main coordinating neurons allow the authors to create a polarized network diagram for these components of the aboral organ. These connections give insight into the potential functions of the major neurons. This also gives some unexpected results, particularly the lack of connections from the balancer system to the coordinating system.

Thank you for these positive comments on the paper.

Weaknesses:There were no significant weaknesses in the paper - only a slate of interesting unanswered questions to motivate future studies.
**Recommendations for the authors:**

**Reviewing Editor Comments:**
In consultation, the reviewers recommend that improving the evidence to “exceptional” would require additional perturbation experiments (e.g., ablation of specific neurons), as Reviewer 1 suggests. They also recommend adding a “Future Directions” section to the manuscript, because it opens up so many new experimental directions.

We have added a new “Future Directions” section at the end of the Discussion. To carry out the proposed perturbation or calcium imaging experiments would require significant additional work and method development. We are actively working in establishing mRNA and DNA injection into ctenophore zygotes to enable live imaging, cell labelling or ablations in the future.

**Reviewer #1 (Recommendations for the authors):**
Suggestions for improved or additional experiments, data, or analyses:To establish causality (neurons control balancer cilia), an important experiment would be to manipulate each of these neuronal populations (e.g., by ablating them) and measure the effect of these ablations on the beating frequency of the balancer cilia of the four quadrants. Moreover, direct observation of neuronal activity (e.g., by using calcium imaging) would also provide more compelling evidence for neuronal control.

We agree with the reviewer that such perturbation experiments would be needed to establish causality. Such experiments are currently still not possible in ctenophoes and would require significant technology development. We discuss such experiments in the “Future directions” section and also place this in the context of the currently available techniques in ctenophores. We are actively working on this but waiting for such technological breakthroughs and new experiments would significantly delay the publication of a version of record of the paper.

Recommendations for improving the writing and presentation:ANN neurons are described in great detail, though SNN neurons are described more loosely. Perhaps a more detailed description of SNN neurons would be helpful.

We added the information on SNNs to show that these cells are distinct from the ANN neurons. Since our focus is on the aboral organ, we did not aim for a comprehensive reconstruction of SNNs. Several of the processes of the SNNs are also truncated and outside our EM volume. We have nevertheless added additional details about the morphology and connectivity of SNN neurons.

“Near the perifery of the aboral organ, we identified four further anastomosing nerve-net neurons. These resembled the previously reported syncytial subepithelial nerve net (SNN) neurons in the body wall of Mnemiopsis (Figure 2–figure supplement 1C–G) and were clearly distinct from the ANN neurons (both in location and morphology). SNN neurons show a blebbed morphology and contain dense core vesicles @Burkhardt2023 but no synapses.”

Minor corrections to the text and figures:(1) (Figure 2 C): “mitochondia” instead of “mitochondria”.

corrected

(2) Figure 3. Title: “balancer and bridge”.

corrected

(3) (Figure 3.C) “shown in xxx color”

corrected

**Reviewer #2 (Recommendations for the authors):**
Clearer usage of the terms statocyst, aboral organ, aboral nerve net, statolith, dome, and lithocytes would be helpful. For readers not familiar with ctenophore anatomy, things can get a bit confusing. A single schematic with all of these terms would be helpful. In Figure 1E, there is a label “dc”. Should this be “do”?

We have added an annotated schematic to Figure 1, explaining these terms.

Figure 1C “The statocyst is a cavity-like organ enclosed by the dome cilia (do), which contains the statolith formed by lithocytes (li) and supported by the balancer cilia (bal).”

**Reviewer #3 (Recommendations for the authors):**
My comments are numerous, but mostly minor suggestions for improving the clarity.[Suggested insertions/changes are indicated by square brackets](1) [It would be much easier to review this if there were line numbers, or with a double-spaced manuscript that was more accommodating for markup.]

Thank you for this comment. We have increased the line spacing in the revised version. (We set the CSS line-height property on the html ‘body’ element to 2em).

(2) The terms statolith, statocyst, and lithocytes can be confusing, so it would be nice to have an upfront definition of how they relate to each other.

We have now explain these terms in the Introduction and also have improved the annotation of Figure 1.

Figure1C. “The statocyst is a cavity-like organ enclosed by the dome cilia (do), which contains the statolith formed by lithocytes (li) and supported by the balancer cilia (bal).”

(3) Statolith is spelled as statolyth in the early pages, but statolith in the later pages. I think -lith is more common, but in any case, these should be standardized.

corrected to ‘statolith’

ABSTRACT:(1) Differential load[s] on the balancer cilia [lead] to altered

changed

(2) We used volume electron microscopy (vEM) to image the aboral organ.

changed

(3) also form reciprocal connections with the bridge cells.

corrected

INTRODUCTION:(1) “identify conserved neuronal markers in ctenophores” - confusing - does this mean conserved across ctenophores, or conserved in ctenophores and other animals?

changed to “classical neuronal markers”

(2) “either increase or decrease their [ciliary] activity, indicating” - otherwise it sounds like the balancers are increasing activity.

changed to “balancer cells may either increase or decrease their ciliary activity”

(3) after “matches the setup used in high-speed imagine experiments”, it might be nice to add a statement like “Future studies could potentially investigate activity in the inverted orientation, when the statolith is suspended below the cilia, to see if the response differs.”

In this sentence we referred to the orientation of the animals in our figures. There is a consensus among ctenophore researchers that when depicting ctenophores, the aboral organ should face downwards. However, for this paper we chose the opposite orientation to better match our experiments and help interpreting the results. We changed the text to: “In this study, we represent ctenophores with their aboral organ facing upwards (”balancer-up” posture), as this configuration facilitates intuitive interpretation of balance-like functions and matches the setup used in high-speed imaging experiments. ”

We added the sentences “Future experiments could also explore how orientation affects the response of balancer cilia. For example, when the statolith is suspended below the cilia (the”balancer-down” posture), ciliary beating patterns may differ from what we observed here in the “balancer-up” configuration.” to the section Future Directions”.

(4) “abolished by calcium[-]channel inhibitors”

corrected

(5) “By functional imaging, we uncovered” - It is not clear what functional imaging is. Maybe a fewword definition here, and be sure to explain in the methods.

changed to “By high-speed ciliary imaging”. The details of the imaging are explained in the Methods section under “Imaging the Activity of Balancer Cilia”.

RESULTS:(1) “five-day-old” - is it worth saying post-fertilization here?

Thank you for pointing this out. In accordance with Presnell et al. (2022), we use post-hatching as the reference. We have revised the text in the Materials and Methods section to read: “5-day-old (5 days post-hatching)”

(2) “We classified these cells into cell types [based on …]” - specify a bit about how you classified them based on morphology, the presence of organelles, etc.

We added a clarification. “Our classification was based on (i) ultrastructural features (e.g. number of cilia), (ii) cell morphology (e.g. nerve net or bridge cells), (iii) unique organelles (e.g. lamellate body, plumose cells), (iv) and similarities to cell types previously described by EM. Our classification agrees with the cell types identified in the 1-day-old larva [@ferraioli2025].”

(3) “CATMAID only supports [bifurcating] skeleton trees” - Correct?

yes, a node in CATMAID cannot be fused to another node of the same skeleton to represent anastomoses

FIGURE 1:(1) It is not worth redrawing and renumbering everything, but I wish the lateral view in A matched the rotated aboral view in B, instead of having to do two rotations to get the alignment to coincide. (Rotating panel B 90{degree sign} clockwise would make them match, but then it wouldn’t coincide with all the subsequent figures.)

Thank you for the suggestion. We have replaced panel A with a lateral view that now matches panel B.

(2) The labels on Figure 1 are a mix of two typefaces (Helvetica and Myriad?). They should be standardized to all use one typeface (preferably Helvetica).

we have changed the font to Helvetica

(3) Panel C legend: arrows are not really arrows. Say “Eye icons” or something like that. Can you show the location of the anal pores in the DIC image?

Changed to ‘eye icons’. The anal pores are usually closed and only open briefly therefore it is not clear where exactly they would be, so indicating their position would be misleading.

(4) Panel F, I cannot see the lines mentioned in the legend at all, except for maybe a tiny wisp in a couple of places. Either omit or make visible.

changed to “The spheres indicate the position of nuclei in the reconstructed cells.”

(5) Panel G. “Cells are color coded according to quadrants”… but unfortunately, the color scale is 90{degree sign} off of what is presented in the rest of the panels and the paper. Q1 and Q3 have been blue, but now Q2+4 are blue/purple, while Q1+3 are orange/yellow. Again, it seems like too much work to recolor panel G, but in future, it would be nice to maintain that consistency, especially since other panels specifically mention the consistent colors.

We have changed the color code in panels B, C and E to match G and the subsequent panels/figures.

RESULTS: Aboral synaptic nerve net(1)“We reconstructed three aboral nerve-net (ANN) neurons” - out of how many total? Were these three just the first ones traced, or are they likely to be all of the multi-domain neurons? One can’t tell if these are the top 3 (out of X), or if there are other multi-quad neurons that were not traced. Are there any Q1Q4 or Q2Q3 neurona? Specify overall composition.

There are only three ANN neurons in the aboral organ. These are all completely reconstructed and contained within the volume. We have clarified this in the text. “We identified and reconstructed three aboral nerve-net (ANN) neurons, each exhibiting a syncytial morphology characterized by anastomosing membranes and multiple nuclei (ranging from two to five) (Figure 2A and B, Figure 2–figure supplement 1C). These three neurons are the only fully reconstructed ANN neurons contained within the volume. Several small ANN-like fragments were also observed at the periphery of the aboral organ, but their connectivity to the main ANN remains uncertain.”

FIGURE 2:(1) Panel C: “N > 2 cells for each cell type” - is that supposed to say “N > 2 mitochondria”? More than 2 cells in all the types shown in the graph.

It is number of cells for each cell type

(2) Panel D: Is this the wrong caption? I can only see green and black circles, not red, yellow, or blue. Make them larger or “flat” (circled, not shaded spheres) if they are supposed to be visible

Thank you for pointing this out. The caption was incorrect and has been corrected to match the figure.

(3) Panel E: Amazing to see the cross-network connections!

Thank you

(4) Again, it is great to see the three ANN mapped out, but … are there other connections that weren’t mapped in this study? Other high-level coordinating neurons? ANN_Q1Q4 or Q2Q3?

The reconstruction is complete and there are no other neurons or connections. Given the large size of ctenophore synapses, we are confident that we identified all or most synapses and their connections.

RESULTS: Synaptic connectome(1) “displaying rotational symmetry” - This is one of the things I am most curious about. Where is the evidence of rotational symmetry in the network diagram? Is it the larger number of connections to Q2 and Q4? Any evidence of rotational symmetry, like Q1 and Q3 connect to Q2 and Q4 respectively, but not the other way around?

changed to “displaying biradial symmetry”, we do not consider the slight difference in synapse number from ANN Q1-4 to the Q1-Q3 vs. Q2-Q4 balancers as significant or strong enough evidence for a single rotational symmetry (i.e. 180 degrees rotation)

(2) “Surprisingly” - this *was* really surprising. There have to be some afferent neurons connecting from the balancers, don’t there? I can’t remember the connections to the SNN, but is there a tertiary set of ANNs that connect between the balancers and the top 3 ANNs? I would like a little more discussion about this.

Indeed, this is why this is so surprising. Most people would have expected some output connections from the balancer to the nerve net or elsewhere. There are none. We have the complete balancer network and all balancer cells are ‘sink nodes’ (inputs only)(Figure3–figure supplement 1).

we added a short statement in the beginning of the Bridge Cells as Feedback Regulators of Ciliary Rhythms section noting that no direct connections from the balancers to the ANN were found and that all balancer cells act as sink nodes (inputs only; Figure 3–figure supplement 1). This highlights that bridge cells are indeed the sole neuronal input to the ANN circuit.

Figure 3:(1) As you know, during development, the diagonally opposite cells have a shared heritage and shared functionality. Are there neuronal signatures that correspond to the rotational symmetry that we see, for example, in the position of the anal pores?

We did not find any evidence in neuronal complement for a diagonal symmetry, suggesting that neuronal organization does not simply mirror the organism’s rotational body symmetry.

(2) Do you have the information to say whether there are any diagonal or asymmetric connections? Can’t tell if those would have shown up in the mapping efforts or if you focused on the major ones only.

Based on our complete mapping, we did not find evidence for a diagonal pattern. The connectivity instead shows a biradial organization.

(3) “extending across opposite quadrant regions” - to me, opposite would be diagonally opposite, but this looks like a set of cells between Q1 and Q2 is connecting to a sister-set in Q3+Q4. I wonder if, in a more detailed view, you could see whether this is a rotational correspondence, rather than a reflection. There are some subtle hints of this in the aboral view, with some cells on the right of the blue cluster and the left of the magenta cluster.

changed to “extending across tentacular-axis-symmetric quadrant regions” for clarity

(4) As with Figure 2, I do not see any circles/spheres that are yellow, red, or blue! There are some traces of what appear to be other neurons that have these colors, but nothing that would suggest the localization of mitochondria.

Thank you for pointing this out. We have corrected the caption to match the figure, as in the previous item.

(5) The connectivity map is very cool, but the caption does not seem to correspond to the version included in the manuscript. I don’t see any hexagons; all arrows seem to have the same thickness.

changed to: “Complete connectivity map of the gravity-sensing neural circuit. Cells belonging to the same group are shown as diamonds, and the number of cells is added to their labels. The number of synapses is shown on the arrows.”

RESULTS: Dynamics of balancer cilia(1) The orientation of the stage+larvae is a bit hard to follow. Maybe say the sagittal or tentacular plane is parallel to the sample stage and the gravity vector?

we added “Larvae were oriented with their sagittal or tentacular plane parallel to the sample stage.”

(2) “We could simultaneously image Q1(3) and Q2(4). The meaning of the numbers in () is not clear. Either way that I try to interpret it does not match the diagrams. Should this say viewing the tentacular plane, you can image Q1 and 4 or Q2 and 3?

Thank you for spotting this mistake, we have changed to: “In larvae with their sagittal plane facing the objective, we could compare balancer-cilia movements between Q1 vs. Q2 or Q3 vs. Q4. In other larvae oriented in the tentacular plane, we could simultaneously image Q1 and Q4 or Q2 and Q3.”

(3) Typo: episod[e]s were excluded

Corrected

DISCUSSION:This section is quite clean. Maybe mention some future directions:

We have added a “Future Directions” section

(1) Do these networks change during development? Five-days-old is still quite undeveloped - what would it look like in an adult specimen? Would you expect a larger version of the same or more diverse connections?

As far as we know from work on aboral organs in adult ctenophores, the same structures and cells can be found. We do not know how the network will develop. We know that at 5 days the balancer is fully functional and the animals can orient and their behaviour is coordinated. So the wiring may not change extensively later in development. In the 1-day-old larva, Ferraioli et al. did not distinguish ANN neurons as a separate population, as these were merged with SNNs in their dataset. This suggests that significant cellular and circuit maturation likely occurs between 1 and 5 days.

METHODS: Imaging the Activity of Balancer Cilia(1) “we selected only larvae whose aboral-oral axis was oriented nearly perpendicular to the gravitational vector”. Shouldn’t this be “nearly parallel to the gravity vector” not perpendicular?

Thank you for spotting this, corrected.